# Hierarchical mechanisms control the clearance of DNA lesion–stalled RNA polymerase II

Paula J. van der Meer [1,5], George Yakoub [1,5], Kotaro Tsukada[2], Yuka Nakazawa [3], Tomoo Ogi [2,4] & Martijn S. Luijsterburg [1]

Stalling of elongating RNA polymerase II (RNAPII) at DNA lesions blocks transcription and triggers transcription-coupled repair (TCR). However, the mechanisms determining the fate of stalled RNAPII remain incompletely understood. Here, we develop a time-resolved assay to track RNAPII clearance and degradation at UV-induced lesions. We show that RNAPII ubiquitylation by CSB and the CRL4$^{CSA}$ ubiquitin ligase is essential, as loss of these proteins causes persistent RNAPII accumulation at damage sites. Downstream of CSB/CRL4$^{CSA}$-mediated ubiquitylation, two distinct pathways mediate RNAPII removal. The primary rapid route relies on TFIIH, with its XPD helicase activity driving RNAPII dissociation after proper recruitment and positioning by ELOF1, UVSSA, and STK19. A secondary slow pathway is mediated by the ubiquitin-dependent segregase VCP, which compensates for impaired TFIIH function. While VCP contributes only minimally in TCR-proficient cells, inhibition of VCP in TFIIH-deficient contexts completely abrogates RNAPII clearance. Together, these findings establish a hierarchical program in which CSB/CRL4$^{CSA}$-mediated ubiquitylation initiates RNAPII processing, TFIIH/XPD helicase activity provides the main clearance mechanism, and VCP-dependent extraction acts as a backup when TFIIH fails. This mechanistic framework explains how cells resolve DNA lesion-stalled RNAPII during normal and compromised TCR.

During transcription, RNA polymerase II (RNAPII) synthesizes nascent RNA as it translocates along the DNA template. However, DNA lesions in the template strand, such as UV-induced photoproducts, stall elongating RNAPII and trigger genome-wide transcriptional arrest[1,2]. Stalled RNAPII functions as a lesion sensor to initiate transcription-coupled repair (TCR), a sub-pathway of nucleotide excision repair (NER) dedicated to removing transcription-blocking lesions from actively transcribed DNA[3–6]. DNA damage-stalled RNAPII is first recognized by the Cockayne syndrome B (CSB) protein, a DNA-dependent ATPase that binds upstream DNA. CSB then recruits CSA, the substrate-recognition subunit of the DDB1-CUL4A-RBX1 ubiquitin ligase complex (CRL4$^{CSA}$)[7–9]. Next, the UV-stimulated scaffold protein A (UVSSA) is recruited in a CSA-dependent manner[9–11]. The transcription elongation factor ELOF1, bound directly to RNAPII, provides a platform for TCR assembly[12,13]. CSA docks onto RNAPII-bound ELOF1, while UVSSA stabilizes the ELOF1-CSA interaction[14]. These steps culminate in ubiquitylation of stalled RNAPII on a conserved lysine (RPB1-K1268)[10,12–15].

[1]Department of Human Genetics, Leiden University Medical Center, Leiden, The Netherlands. [2]Department of Genetics, Research Institute of Environmental Medicine, Nagoya University, Nagoya, Japan. [3]Department of Molecular Genetics, Center for Neurological Diseases and Cancer, Nagoya University Graduate School of Medicine, Nagoya, Japan. [4]Department of Human Genetics, Nagoya University Graduate School of Medicine, Nagoya, Japan. [5]These authors contributed equally: Paula J. van der Meer, George Yakoub. ✉e-mail: m.luijsterburg@lumc.nl

RNAPII ubiquitylation enables recruitment of transcription factor IIH (TFIIH)[9], a core NER factor required in both TCR and global genome repair (GGR). In GGR, the damage sensor XPC recruits TFIIH[16,17], whereas in TCR, UVSSA recruits TFIIH through a flexible TFIIH-interacting region[12,14,18]. The recruitment of TFIIH additionally requires RNAPII ubiquitylation, while its transfer to RNAPII is mediated by UVSSA ubiquitylation[10]. Subsequently, the recently identified TCR factor STK19[19–22] positions TFIIH in front of RNAPII via an interaction with the XPD helicase subunit. TFIIH then recruits XPA and the endo-nucleases XPG and ERCC1-XPF, leading to DNA unwinding and dual incision around the lesion[23–27]. The excised 26–28 nucleotide DNA fragment containing the lesion is subsequently replaced through repair synthesis, and the resulting gap is sealed by ligation[27–29].

TCR factor assembly and RNAPII ubiquitylation ultimately lead to the displacement of RNAPII, which otherwise blocks lesion access to downstream repair proteins[5]. During this process, RNAPII undergoes three sequential steps: (i) ubiquitylation of the largest subunit RPB1 at K1268, (ii) removal of RNAPII from the DNA lesion, and (iii) proteasome-mediated degradation of ubiquitylated RNAPII. The regulation and fate of lesion-stalled RNAPII during these steps have remained unclear, largely due to the lack of direct experimental tools[1,6,30,31]. Recently, two complementary methods were developed to study RNAPII clearance: fluorescence recovery after photobleaching (FRAP) in live cells[32], and the sequencing-based method, PADD-seq[33]. These revealed that persistent RNAPII stalling occurs selectively in CSB- or CSA-deficient, but not UVSSA-deficient cells. Earlier work similarly showed that CSB- or CSA-deficient fibroblasts are impaired in RNAPII ubiquitylation and degradation after UV[34,35]. However, a confounding factor is that CSB or CSA loss disrupts all downstream TCR steps, including UVSSA, STK19, and TFIIH recruitment, making it difficult to assign specific roles in RNAPII clearance[9,12,14,19]. Notably, UVSSA-deficient cells exhibit reduced levels of RNAPII ubiquitylation, yet retain the capacity for RNAPII removal and degradation[14,35]. In contrast, STK19-deficient cells display prolonged retention of ubiquitylated RNAPII and delayed polymerase clearance[19]. These observations suggest that ubiquitylation is necessary but not sufficient for the proper clearance of RNAPII.

To systematically dissect these mechanisms, we developed a new, straightforward imaging-based assay to directly measure the clearance of DNA damage–stalled RNAPII. We applied this approach to a panel of eight isogenic knockout cell lines, each lacking a distinct TCR factor, as well as separation-of-function mutants: (i) a CSA mutant that permits RNAPII ubiquitylation but fails to recruit UVSSA[14], (ii) a UVSSA mutant that supports ubiquitylation but fails to recruit TFIIH[14], and (iii) helicase-dead XPD mutants that support transcription but not repair[36]. We also tested the contribution of the ubiquitin-selective segregase VCP/p97[37]. Our findings support a hierarchical model in which CSB/CRL4$^{CSA}$-dependent ubiquitylation primes RNAPII for clearance, with TFIIH-driven helicase activity acting as the primary rapid pathway, and VCP-dependent extraction serving as a slower backup when TFIIH activity is impaired.

## Results

### Local DRB run-off: a method to track DNA damage-stalled RNAPII clearance in situ

Our understanding of the fate of RNAPII stalled at DNA lesions remains incomplete because we lack methods to visualize this process in single cells in a spatiotemporal manner. Therefore, we developed an assay to induce a single region of local UV damage per nucleus using micropore filters[38], then tracked the fate of RNAPII by measuring its clearance at the damage site (Fig. 1a). To specifically measure the levels of already elongating RNAPII without interference from newly initiated RNAPII molecules that might stall at unrepaired lesions, we treated cells with the reversible transcription elongation inhibitor 5,6-dichloro-1-β-D-ribofuranosylbenzimidazole (DRB) immediately after UV irradiation.

DRB prevents new RNAPII complexes from transitioning from the promoter-proximal pause site (marked by RNAPII-S5P) to the hyper-phosphorylated elongating form (RNAPIIo, containing both RNAPII-S2P and RNAPII-S5P) (Supplementary Fig. 1a). The clearance of damage-stalled RNAPII from the damaged area was monitored over time using immunofluorescence (IF) staining of elongating RNAPII-S2P (Fig. 1a). By applying this assay to RPE1-hTERT cells, we observed a gradual decrease in RNAPII-S2P during a 2-hour time-course after DRB treatment, outside the locally damaged area in the nucleus (Fig. 1b). This reduction is consistent with the notion that elongating RNAPII complexes (travelling ~120 kb in 1 h) eventually run off gene bodies and become dephosphorylated. However, at the locally inflicted sites of UV damage, marked by staining of cyclobutane pyrimidine dimers (CPD), we observed an accelerated loss ( ~ 2-fold) of RNAPII-S2P compared to the signal outside of the damaged area, reflecting the active clearance of DNA damage-stalled RNAPII from chromatin (Fig. 1b-c). Thus, our new assay directly measures the clearance of stalled RNAPII from DNA lesions.

### Global DRB run-off: a method to detect DNA damage-stalled RNAPII degradation

Although the local DRB run-off assay can measure the clearance of elongating RNAPII from UV lesions, it falls short in determining the fate of RNAPII after its eviction. Therefore, we modified our local DRB run-off approach to investigate RNAPII degradation. To this end, cells were globally exposed to UV and incubated in DRB-containing media. After 1 and 2-hour intervals, the levels of RNAPII-S2P and total RNAPII (detecting both initiating RNAPIIa and elongating RNAPIIo) were analyzed from whole cell extracts by SDS-PAGE and western blotting (Fig. 1d). This approach reflects the levels of chromatin-associated elongating RNAPII-S2P, as RNAPII phosphorylation occurs only on chromatin. Similar to the local IF approach, DRB treatment alone without UV irradiation caused a gradual reduction in RNAPII-S2P levels, indicating that elongating RNAPII ran off gene bodies and became de-phosphorylated (Fig. 1e-f). Interestingly, UV-irradiated samples after DRB showed a faster reduction in RNAPII-S2P compared to unirradiated samples treated with DRB, indicating active degradation upon eviction from DNA lesions (Fig. 1e-g). Staining with an RNAPII-S5P antibody revealed the loss of a high molecular weight band after UV irradiation, corresponding to the elongating, hyper-phosphorylated RNAPIIo form. In contrast, a lower RNAPII-S5P band persisted, migrating between the unphosphorylated (RNAPIIa) and hyperphosphorylated (RNAPIIo) bands, and representing promoter-proximally paused RNAPII, which persists when cells were treated with DRB (Supplementary Fig. 1a). Over the past two decades, studies have shown that damage-stalled RNAPII undergoes ubiquitylation and proteasome-mediated degradation, a process inhibited by MG132[32,33,35,39]. Therefore, we included the proteasome inhibitor MG132 in our experiments. Consistent with previous reports, both local and global DRB run-off assays showed that proteasome inhibition abrogated RNAPII clearance and degradation (Fig. 1h-k). However, MG132 can also deplete the free ubiquitin pool, thereby impairing ubiquitylation more broadly[40]. Accordingly, immunoprecipitation of RNAPII-S2P after MG132 treatment revealed a marked reduction in RNAPII-S2P ubiquitylation (Supplementary Fig. 1b), indicating that the observed defects in RNAPII-S2P clearance and degradation likely reflect combined effects of proteasome inhibition and impaired ubiquitylation. Taken together, we established two parallel approaches to specifically monitor the clearance and degradation of elongating RNAPII upon encountering UV-induced DNA damage.

### Validating an isogenic collection of TCR-defective cell lines

To investigate the role of individual TCR factors in the active clearance and degradation of lesion-stalled RNAPII, we utilized a collection of eight isogenic TCR-gene knockouts (KO) in human RPE1-hTERT cells.

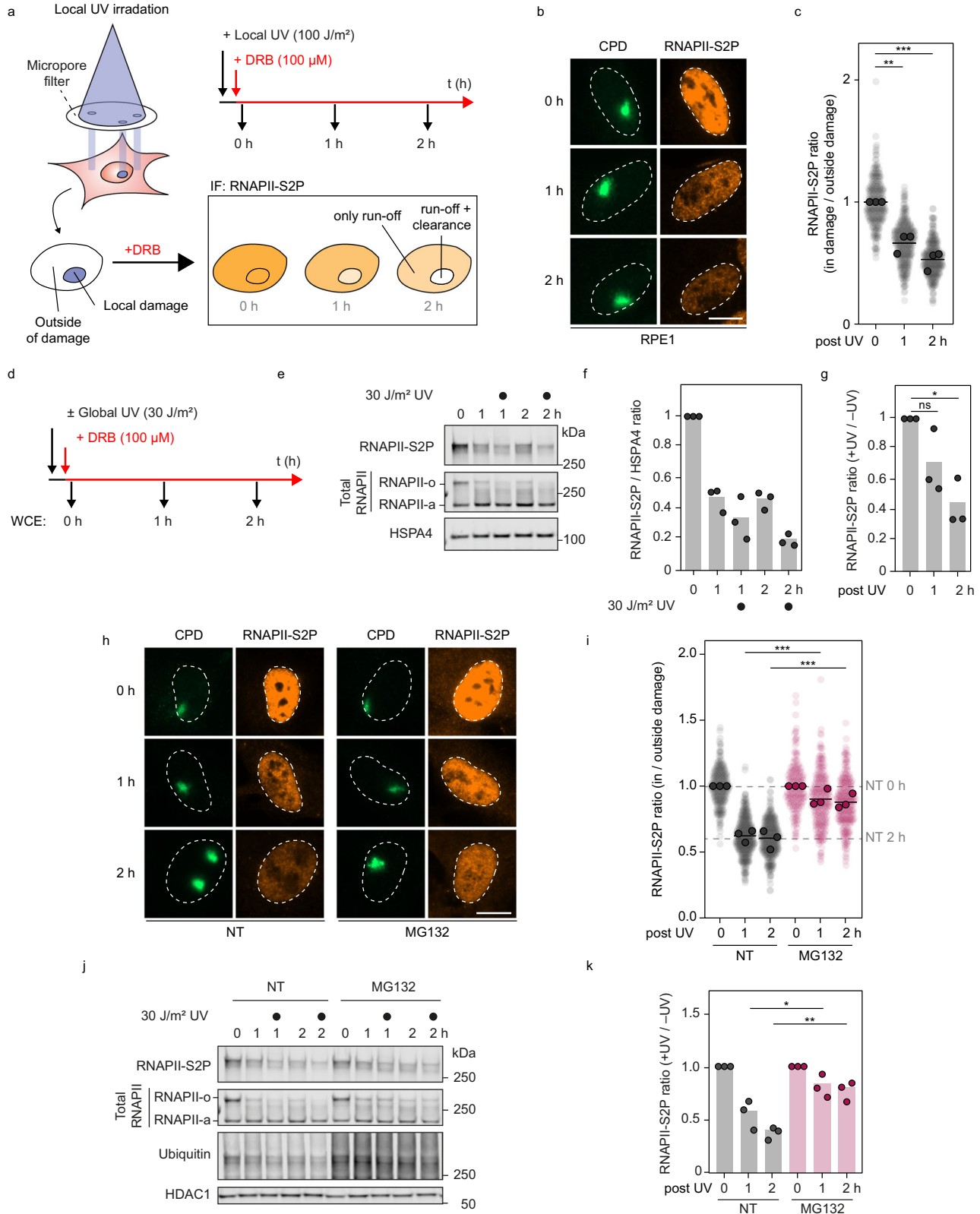

This collection includes cells with genetic inactivation of early TCR factors (CSB, CSA), factors that stabilize the RNAPII-bound TCR complex and recruit or position TFIIH (ELOF1, UVSSA, STK19), and downstream NER factors that bind TFIIH and promote excision of the lesion (XPA, XPG, ERCC1)[9,19,41–43]. The genetic deletion of most TCR genes was confirmed by western blot analysis, showing the loss of the respective TCR proteins (Supplementary Fig. 1c). Knockouts of ELOF1, UVSSA,

and STK19 were validated by Sanger sequencing because commercially available antibodies do not recognize these TCR proteins in RPE1-hTERT cells (Supplementary Fig. 1 d).

A hallmark of TCR deficiency is the inability to recover transcription after DNA damage, such as UV irradiation. To confirm that all cell lines in our collection are deficient in TCR, we measured transcription recovery in each case. To this end, we metabolically labeled

**Fig. 1 | Experimental approach to measure local and global DRB run-off of elongating RNAPII. a** Schematic of local DRB run-off assay by immuno-fluorescence imaging. Cells were treated with 100 µM DRB, preventing new RNAPII molecules from going into elongation, immediately after local UV-C (100 J/m2), then fixed after 0, 1, or 2 h and stained by immunofluorescence. **b** Representative images of (**a**), stained for RNAPII-S2P (Abcam, Ab5095) in RPE1 WT cells at the indicated time points after local UV-C, marked by CPD (Cosmo Bio, CAC-NM-DND-001). Scale bar, 10 µm. **c** Quantification of (**b**). RNAPII-S2P mean pixel intensity inside local damage was divided by that outside local damage in the same cells to correct for the general RNAPII run-off caused by DRB. The experiment was performed three times ($n = 3$). Each transparent colored circle represents one cell, each outlined solid circle the mean of two technical replicates ( > 50 cells / technical replicate). Black lines represent the means of three independent experiments. **d** Schematic of global DRB run-off assay by western blotting. RNAPII was detected by western blot from whole cell lysates of RPE1 WT cells at 0, 1, and 2 h after global

UV-C (30 J/m²) and incubation with 100 µM DRB. **e** Representative image of (**d**). The antibody against RNAPII-S2P (Millipore, 04-1571 (3E10)) detects only elongating RNAPII (upper blot). The antibody against the N-terminal domain of RNAPII (Cell Signaling, 14958 (D8L4Y)) detects both initiating (RNAPIIa) and elongating (RNAPIIo) forms (middle blot). HSPA4 was used as loading control. **f** Quantification of blots in (**e**) RNAPII-S2P mean pixel intensity was divided by that of HSPA4 and normalized to 0 h. **g** RNAPII-S2P + UV / -UV ratio of quantification in (**f**). The value of each UV-treated time point in (**f**) was normalized to its respective unirradiated condition. **h** Representative images of local DRB run-off with and without proteasome inhibitor (MG132; 20 µM; 1 h pretreatment and the indicated incubation times after UV-C). **i** Quantification of (**h**) as described in (**c**). **j** Representative image of global DRB run-off assay with and without proteasome inhibitor as described for (**h**). HDAC1 was used as loading control. RNAPII antibodies are as in (**e**). **k** Quantification of (**j**) as described in (**f**) and (**g**).

and visualized nascent transcripts by 5-ethynyl-2-uridine (EU) incorporation, followed by microscopy. All KO cell lines were defective in transcription recovery after UV irradiation (Fig. 2a-b, Supplementary Fig. 2a). To further demonstrate impaired TCR in each KO line, we treated cells with the chemotherapeutic trabectedin, which generates RNAPII-stalling lesions that are processed by TCR[14,19,44]. This processing creates persistent single-strand breaks that are converted into double-strand breaks (DSBs) during replication[45] (Supplementary Fig. 2b). Importantly, these DSBs are not generated when TCR is defective[14]. Thus, we treated cells with trabectedin and measured γH2AX levels in replicating (EdU-positive) cells after 4 h. WT cells showed elevated γH2AX levels upon trabectedin treatment, consistent with proficient TCR (Supplementary Fig. 2c-d). In contrast, all eight KO lines had severely reduced γH2AX levels, confirming their TCR defect. This unique isogenic KO collection enables direct side-by-side comparison of how different factors at various stages of TCR affect RNAPII clearance and degradation.

**The impact of TCR factors on the clearance and degradation of lesion-stalled RNAPII**

We next used the local DRB run-off approach to track RNAPII clearance in our isogenic collection of TCR knockouts (Fig. 2c-d, Supplementary Fig. 3a). Due to the DRB treatment, the levels of elongating RNAPII outside of the locally damaged area in the nucleus decreased gradually in each cell line over a time course of 2 h, as observed before (Fig. 1a-c). While WT cells displayed an accelerated loss of RNAPII-S2P signal at the sites of local UV damage, the ratio between RNAPII-S2P signal inside and outside of the local damage in CSB[KO] and CSA[KO] cells stayed high, reflecting a defect in the removal of stalled RNAPII-S2P from DNA lesions (Fig. 2c-d, Supplementary Fig. 3a). In ELOF1[KO], UVSSA[KO,] and STK19[KO] cells, the removal of damage-stalled RNAPII was attenuated at 1 h after UV irradiation, exhibiting a phenotype intermediate between that of WT and CSB[KO] or CSA[KO] cells. However, 2 h after UV treatment, RNAPII was largely cleared in these TCR KOs. Interestingly, the clearance of damage-stalled RNAPII in XPA[KO], XPG[KO], and ERCC1[KO] cells was not significantly different from that in WT cells (Fig. 2c-d, Supplementary Fig. 3a). Hence, based on our new assay, we can categorize the TCR factors into three groups: (i) CSB[ko] and CSA[KO] showing a strong clearance defect, (ii) ELOF1[KO], UVSSA[KO,] and STK19[KO] cells showing delayed polymerase removal, and (iii) XPA[KO], XPG[KO], and ERCC1[KO] showing no impact on RNAPII removal.

To assess whether the observed clearance phenotypes in the TCR KO cell lines culminate in the degradation of elongating RNAPII after UV irradiation, we repeated the global DRB run-off assay in the eight isogenic TCR knockout collection. The cells were globally exposed to UV damage and incubated in DRB-containing media. Similar to the local imaging approach, DRB treatment alone, without any UV irradiation, caused a gradual reduction in hyperphosphorylated elongating RNAPII-S2P levels in all tested TCR knockouts, suggesting that

elongating RNAPIIo ran off gene bodies, while unphosphorylated promoter-bound RNAPIIa was still present (Fig. 2e-f, Supplementary Fig. 3b-c). In contrast, UV-treated samples showed a faster reduction of RNAPII-S2P in WT cells, suggesting its active degradation due to TCR. Interestingly, only CSB[KO] and CSA[KO] cells, but none of the other TCR knockouts, displayed significantly impaired degradation of RNAPII-S2P after UV damage (Fig. 2e-f, Supplementary Fig. 3b-c). We suspect that this assay lacks the sensitivity to detect a delayed phenotype at 1 h after UV irradiation in ELOF1[KO], UVSSA[KO,] and STK19[KO] cells, unlike the imaging-based clearance assay, which has a broader dynamic range and can reveal more subtle differences.

**UV-induced ubiquitylation of RNAPII is required for RNAPII processing**

We next wondered why CSB[KO] and CSA[KO] cells fail to clear and subsequently degrade damage-stalled RNAPII, whereas the other TCR KO cell lines do not. Upon stalling on a lesion, RNAPII's largest subunit is ubiquitylated on a conserved lysine (RPB1[K1268]) by CSB-mediated recruitment of the CRL4[CSA] E3 ligase complex and subsequent docking on ELOF1, which is further stabilized by UVSSA to direct the ubiquitylation[10,14]. This TCR-associated ubiquitylation is essential for efficient damage repair and RNAPII degradation following UV irradiation[10,12,15]. To examine side-by-side to what extent the TCR-defective cell lines are proficient in UV-induced ubiquitylation of RNAPII, we irradiated cells and analyzed the ubiquitylation of RNAPII after immunoprecipitation of RPB1-S2P. We observed a ubiquitin signal at the expected height of Ub-RPB1 in WT cells upon UV irradiation, indicating efficient RNAPII ubiquitylation. Consistent with their known role in RNAPII ubiquitylation, CSB[KO] and CSA[KO] cells did not show any detectable RPB1 ubiquitylation, reflecting their defects in RNAPII clearance and degradation at UV damage sites (Fig. 3a). Intriguingly, RPB1 ubiquitylation was strongly reduced but not completely abolished in ELOF1[KO] and UVSSA[KO], correlating with their slower, but functional RNAPII clearance rate. In STK19[KO], which also showed a delayed RNAPII clearance, RPB1 ubiquitylation was comparable to that of the WT cells. KO cell lines of the downstream TCR factors (XPA[KO], XPG[KO], and ERCC1[KO]) showed no defect in RPB1 ubiquitylation (Fig. 3a).

Activation of CRL4[CSA] requires conjugation of the ubiquitin-like protein NEDD8[46]. Thus, inhibition of CRL4 neddylation prevents RNAPII ubiquitylation[10,14,15]. To substantiate that the lack of RNAPII ubiquitylation causes the clearance defects in CSB[KO] and CSA[KO] cells, we inhibited CRL activity with the neddylation inhibitor (MLN4924), which prevents RNAPII ubiquitylation by the CRL4[CSA] E3 ligase. RNAPII clearance and degradation under these conditions were both entirely defective in our local and global DRB run-off approaches (Fig. 3b-d). To further test the idea that the ubiquitylation function of CSA is important for RNAPII clearance, we stably expressed a mutant CSA protein in CSA[KO] cells, which does not interact with the CRL4[CSA] E3 ligase complex (CSA[ΔCRL4])[47]. Re-expression of CSA[WT] fully restored RNAPII clearance in

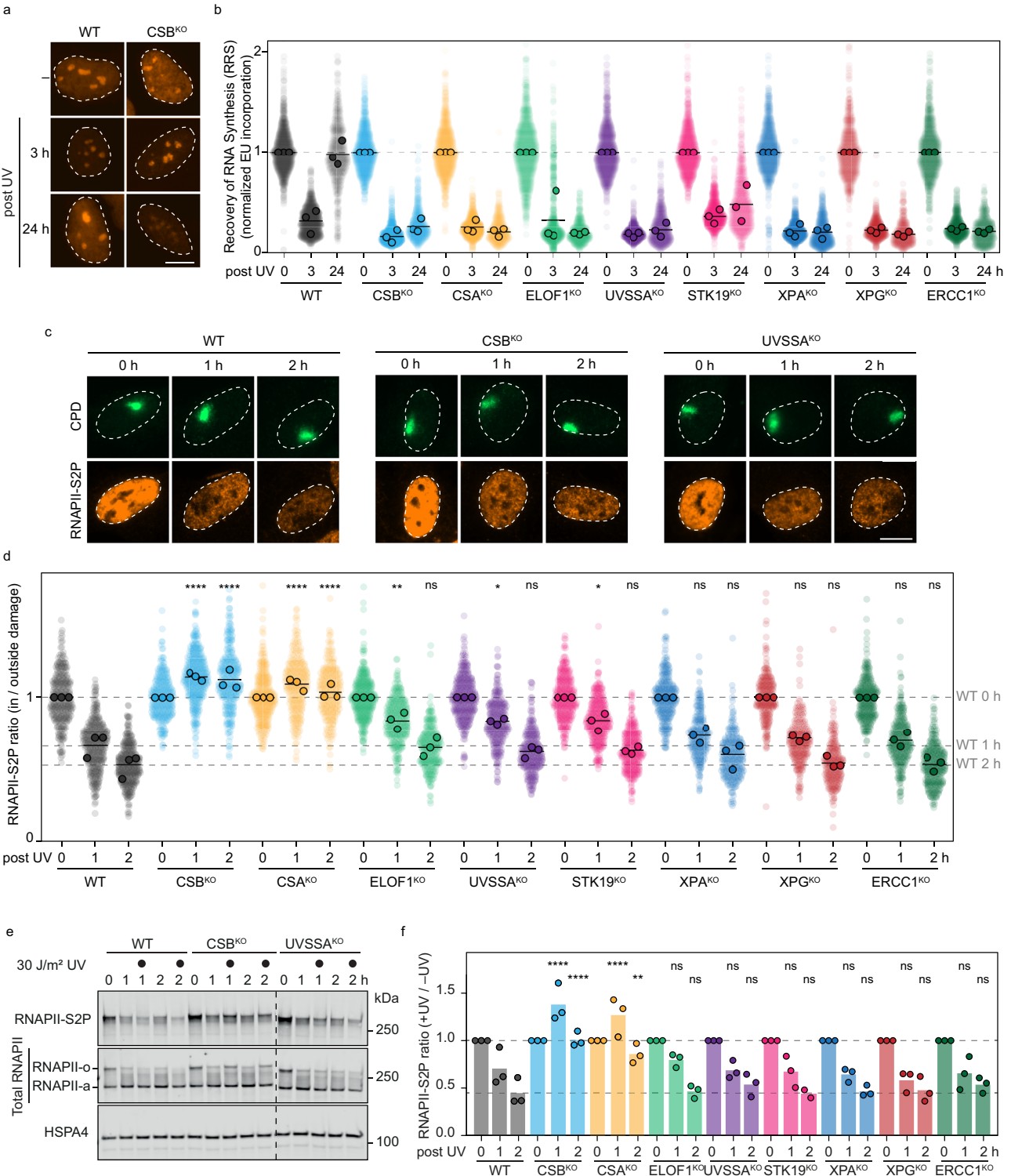

**Fig. 2 | TCR activity, RNAPII clearance and RNAPII degradation in an isogenic collection of TCR knockout cells. a** Representative images of recovery of RNA synthesis (RRS), measured by incorporation of 5-EU (1 h labeling), either mock-treated (-UV) or 3 h or 24 h after UV-C irradiation (9 J/m²). Only WT and CSB^KO are shown here. See Supplementary Fig. 2a for the other conditions. **b** Quantification of RRS as in (**a**). The nascent transcription levels of the indicated cell lines, where 5-EU levels were normalized to the mock treatment for each cell line. The experiment was performed three times. Each transparent colored circle represents one cell. Each dark circle represents the mean of two technical replicates, with more than 50 cells collected per technical replicate. The black lines represent the mean of all three independent experiments. **c** Representative images of the local DRB run-off assay, as in Fig. 1a-b in WT, CSB^KO, and UVSSA^KO cells. See Supplementary Fig. 3a for representative images of the other conditions. Scale bar, 10 μm. **d** Quantification of (**c**) for the indicated RPE1 WT and TCR^KO cells as described for Fig. 1c. **e** A representative image of the global DRB run-off assay by western blotting in the indicated RPE1 WT and TCR^KO cells as described for Fig. 1e. The dashed line indicates the merge zone of cropped blots. **f** Quantification of (**e**) as described for Fig. 1f-g.

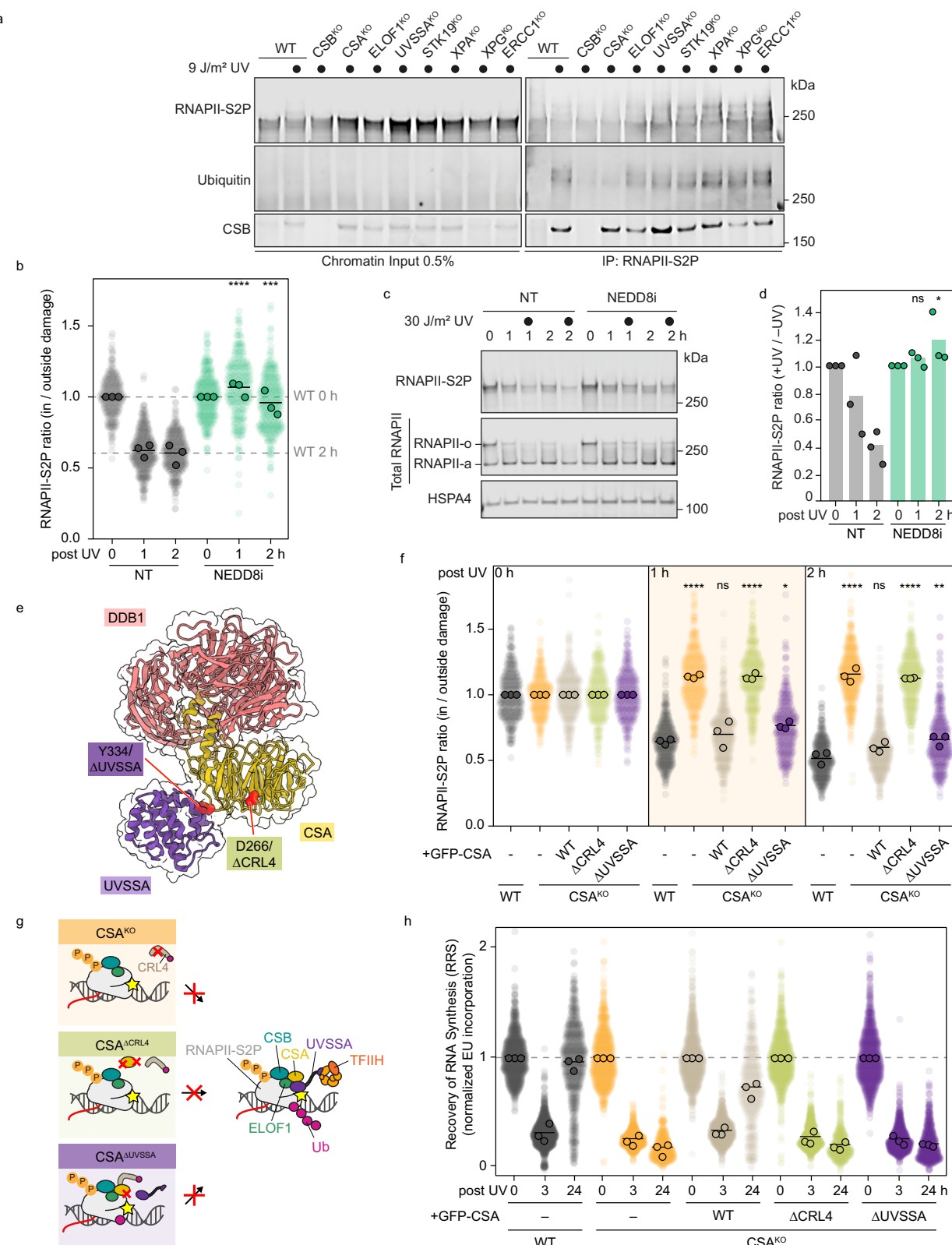

**Fig. 3 | Ubiquitylation of damage-stalled RNAPII is required for its clearance.**
**a** Detection of RNAPII-S2P ubiquitylation detected by western blot after immuno-precipitation of RNAPII-S2P (right) from the chromatin fraction (left) of RPE1 WT cells and the indicated TCR knockouts. Cells were treated with 12 J/m² UV-C and collected after 1 h. **b** Quantification of the local DRB run-off immunofluorescence assay as described for Fig. 1a-c with and without NEDD8 inhibitor treatment (MLN4924; 10 μM; 1 h pretreatment and then for the indicated incubation times after UV-C treatment). **c** A representative image of the global DRB run-off assay by western blotting as in Fig. 1c, with and without NEDD8 inhibitor treatment.

**d** Quantification of (**c**), as described in Fig. 1f-g. **e** Protein structures of CSA interacting with UVSSA and DDB1, with two amino acids highlighted that are required for the general CRL4 stability (D266) and UVSSA recruitment (Y334). Structures are from Kokic et al. 2024 (PDB: 8B3D)[14]. **f** Quantification of the local DRB run-off immunofluorescence assay as described for Fig. 1a-c for 0, 1, and 2 h after local UV-C treatment in RPE1 WT, CSA^KO, and CSA^KO stably expressing the indicated CSA constructs. **g** Cartoon depicting the effect of CSA^KO or complemented CSA mutants on TCR complex assembly. **h.** Quantification of RRS in RPE1 WT, CSA^KO, and CSA^KO stably expressing the indicated CSA constructs as described in Fig. 2a-b.

CSA[KO] cells, whereas expression of the CSA[ΔCRL4] mutant still resulted in a full clearance defect. In contrast, stable expression of a CSA mutant protein that interacts with CRL4 but fails to interact with UVSSA (CSA[ΔUVSSA])[14] led to a restoration of the clearance phenotype to the attenuated clearance observed in UVSSA[KO] cells (Fig. 3e-g, Supplementary Fig. 4a-c). Importantly, both CSA mutants were equally defective in TCR as measured by transcription recovery assays (Fig. 3g-h). Taken together, these data indicate that ubiquitylation of RNAPII is a crucial step for the clearance and degradation of RNAPII during TCR. Without RNAPII ubiquitylation, stalled RNAPII persists on the lesion and is not degraded, whereas reduced ubiquitylation results in attenuated clearance.

### TCR-defective primary cells confirm RNAPII ubiquitylation and degradation defects

To determine whether the insights from our new assays are specific to isogenic RPE1-hTERT cell lines or reproducible across different genetic backgrounds, we extended our global UV-damage experiments to analyze RNAPII ubiquitylation and degradation in a panel of primary fibroblast lines derived from patients with TCR-deficient genodermatoses. To this end, we treated seven fibroblast lines, along with a WT control, with cycloheximide to inhibit the synthesis of new RPB1 to better visualize RNAPII ubiquitylation and degradation profiles[35]. Using an antibody recognizing elongating RNAPIIo, we detected RNAPII ubiquitylation and degradation during a time-course of 6 h in WT fibroblasts (Supplementary Fig. 4 d). Primary fibroblasts from individuals with CSB or CSA deficiency showed reduced RNAPII ubiquitylation and impaired degradation of RPB1. While RNAPII ubiquitylation was attenuated in primary fibroblasts with a UVSSA defect, we still detected RNAPII degradation, as we previously reported[35]. In contrast, primary fibroblasts with deficiencies in XPA, XPG, ERCC1, or XPF all displayed normal ubiquitylation and degradation of RNAPII (Supplementary Fig. 4 d). The results obtained using primary fibroblasts align well with our findings using isogenic TCR knockout cells, revealing that the ubiquitylation and degradation of DNA damage-stalled RNAPII are selectively impaired in CSB and CSA-defective cells. Notably, UVSSA-defective primary and RPE1-hTERT cells exhibit reduced ubiquitylation yet retain the ability to degrade RNAPII. These results suggest that low levels of RNAPII ubiquitylation may still trigger degradation by alternative mechanisms.

### Under-ubiquitylated RNAPII is cleared by the VCP segregase in UVSSA-deficient cells

To further assess how lesion-stalled RNAPII may be removed and degraded in UVSSA-deficient cells despite low levels of RNAPII ubiquitylation, we considered analyzing the involvement of the ubiquitin-dependent segregase VCP, also known as p97[37]. VCP extracts chromatin-bound ubiquitylated proteins and shuttles them to the proteasome, and it has been implicated in RNAPII clearance during TCR[32,33]. To investigate VCP's contribution to the clearance of damage-stalled and ubiquitylated RNAPII, we examined the effect of a specific VCP inhibitor (VCPi) on RNAPII removal. Treatment of cells with VCPi led to a small but nonsignificant effect on RNAPII clearance in WT cells, indicating that VCP-mediated chromatin extraction does not play a significant role in TCR-proficient cells (Fig. 4a, Supplementary Fig. 5a). In contrast, RNAPII clearance became fully dependent on VCP in both UVSSA[KO] cells (Fig. 4a, Supplementary Fig. 5a) and UVSSA-defective primary fibroblasts (Fig. 4b). These results align well with the proposed role of VCP in repair-independent eviction of lesion-stalled RNAPII in the absence of UVSSA, detected by Damage-seq and PADD-seq[33]. While UV-induced RNAPII ubiquitylation was barely detectable in UVSSA[KO] cells after RNAPII-S2P pull-down, we observed a marked increase when cells were treated with VCPi (Figs. 3a and 4c), suggesting a VCP-dependent clearance and proteosome-dependent degradation of ubiquitylated RNAPII in TCR-deficient UVSSA[KO] cells.

UVSSA contributes to RNAPII ubiquitylation by stabilizing the docking of CSA onto RNAPII-bound ELOF1, which positions CRL4[CSA] on the surface of RNAPII[14]. Therefore, ELOF1[KO] cells also show reduced levels of ubiquitylated RNAPII after UV (Fig. 3a). Similar to UVSSA[KO] cells, we found that RNAPII clearance in ELOF1[KO] cells was also largely dependent on VCP-mediated extraction (Supplementary Fig. 5b). These data show that while VCP is dispensable for RNAPII removal in TCR-proficient cells, it becomes essential for clearing under-ubiquitylated RNAPII resulting from misassembled TCR complexes in both UVSSA- and ELOF1-deficient cells.

### VCP-dependent RNAPII clearance compensates for compromised TFIIH function

We next wondered whether the need for VCP-dependent clearance in the absence of UVSSA and ELOF1 is caused by inefficient RNAPII ubiquitylation or the lack of TFIIH recruitment, which are both phenotypes of UVSSA[KO] and ELOF1[KO] cells[14]. The N-terminal VHS domain of UVSSA stabilizes the CSA-ELOF1 interface and thereby promotes efficient RNAPII ubiquitylation, whereas a more C-terminally located Zn-finger domain docks onto the jaw of RNAPII and facilitates TFIIH recruitment[14]. To distinguish between these separate functions of UVSSA, we stably expressed a UVSSA Zn-finger mutant[14] that rescues RNAPII ubiquitylation to normal levels, but is defective in TFIIH recruitment (UVSSA[ΔTFIIH]) (Fig. 4d). Intriguingly, bringing back only ubiquitylation but not TFIIH recruitment did not rescue the delayed RNAPII clearance observed at 1 h after UV in UVSSA[KO] cells to the WT level (Fig. 4e, left; Supplementary Fig. 5c, left). Moreover, treatment with VCPi revealed that RNAPII clearance in UVSSA[ΔTFIIH] cells is largely dependent on VCP (Fig. 4e, right, Supplementary Fig. 5c, right). These results reveal an important contribution of TFIIH to RNAPII clearance, which is partially masked by a compensatory VCP-dependent RNAPII clearance pathway. Moreover, this separation-of-function mutant shows that it is not the lack of RNAPII ubiquitylation, but the lack of TFIIH recruitment that triggers the VCP-dependent clearance of RNAPII. The situation with the UVSSA[ΔTFIIH] mutant with normal RNAPII ubiquitylation but defective TFIIH recruitment is similar to what happens in STK19[KO] cells. In STK19-deficient cells, RNAPII ubiquitylation occurs normally (Fig. 3a) and TFIIH is recruited by UVSSA, but TFIIH is mispositioned on RNAPII[19,20,22]. Importantly, RNAPII clearance in STK19[KO] cells was also largely dependent on VCP (Fig. 4f, Supplementary Fig. 5 d), suggesting that TFIIH mispositioning also triggers VCP-dependent RNAPII clearance.

### XPD helicase activity facilitates RNAPII clearance during proficient TCR

Under conditions of normal RNAPII ubiquitylation, both TFIIH recruitment by UVSSA[9,14] and TFIIH positioning by STK19[19,20] contribute to efficient RNAPII clearance. UVSSA initially recruits TFIIH via its p62 subunit, after which STK19 interacts with XPD to correctly position it in front of RNAPII[5]. XPD is an ATP-dependent 5' to 3' helicase whose activity is dispensable for transcription initiation and does not engage DNA during this process[48]. Structural rearrangements within TFIIH activate XPD for DNA repair by enabling DNA binding and lesion verification[36]. During TCR, the lesion resides in RNAPII's active site, with XPD likely bound to the downstream DNA in front of the polymerase. XPD's helicase activity could therefore destabilize RNAPII, leading to its dissociation. To test the impact of inactive XPD on RNAPII clearance, we analyzed a primary human fibroblast line expressing a helicase-dead XPD mutant (XPD[G675R]) that causes xeroderma pigmentosum combined with Cockayne syndrome[49] (Fig. 5a, Supplementary Fig. 6a). The mutation was confirmed by Sanger sequencing (Supplementary Fig. 6a), and expression of the mutant protein was comparable to WT levels (Fig. 5a). As expected, these cells were fully defective in TCR, as indicated by their inability to recover transcription after UV irradiation (Fig. 5b).

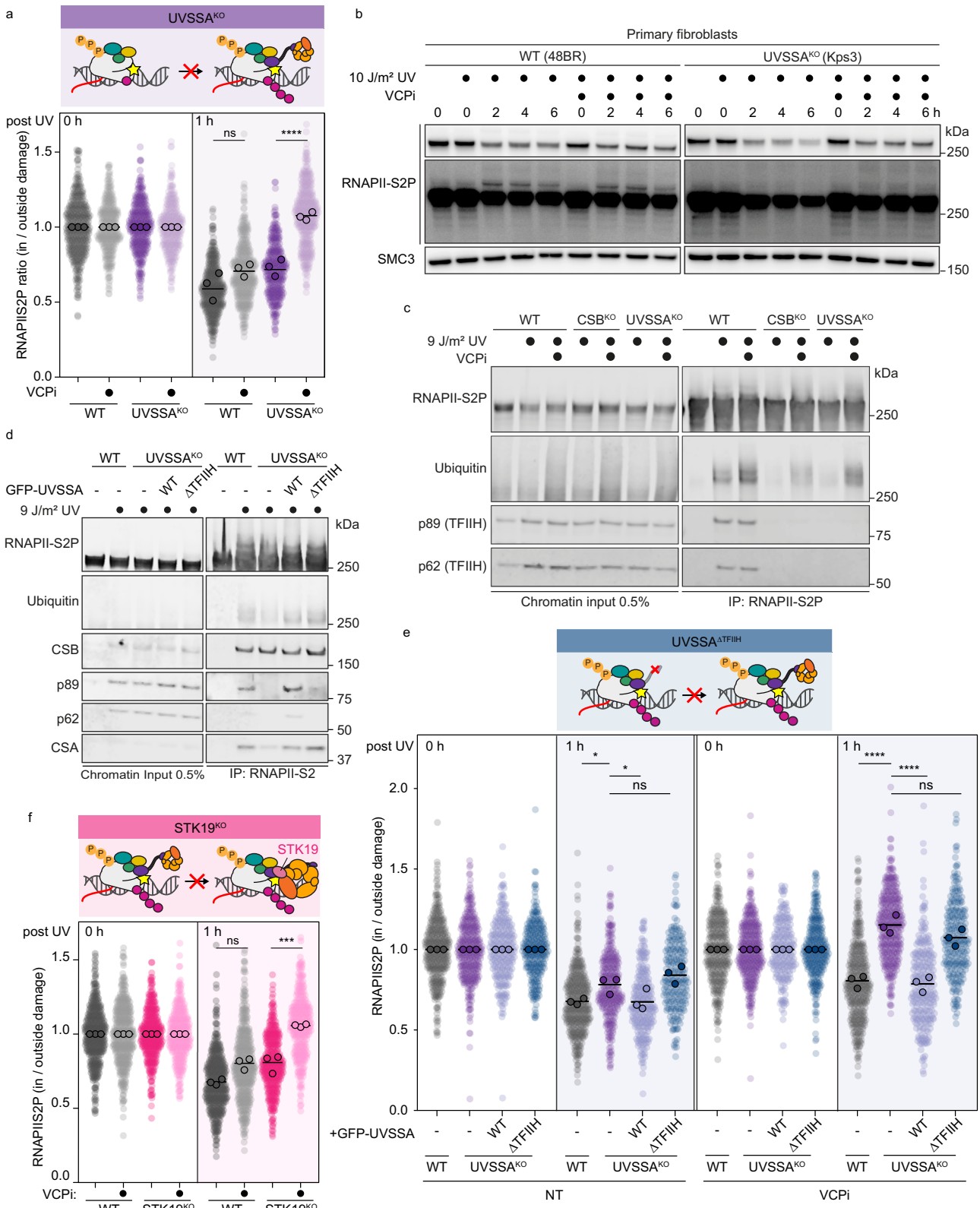

Using the global DRB run-off approach, we observed markedly slower degradation of RNAPII following UV irradiation in helicase-dead XPD cells compared to TCR-proficient primary fibroblasts (Fig. 5c-d). In the local RNAPII clearance assay, XPD^G675R cells showed a pronounced delay in RNAPII removal from sites of local damage. This delay was specifically due to inactive XPD, as preventing TFIIH recruitment via siRNA-mediated knockdown of UVSSA reduced the severe clearance defect to the milder delay previously observed in UVSSA^KO cells (Fig. 5e). The clearance defect observed in XPD^G675R cells was more pronounced than that seen in UVSSA^KO cells, but still milder than the defective removal detected in CSB^KO and CSA^KO. We therefore tested whether the residual clearance in the XPD helicase-dead cells was also facilitated by the VCP segregase. Indeed, VCPi treatment fully abolished the residual RNAPII clearance in primary XPD^G675R cells (Fig. 5f).

**Fig. 4 | Under-ubiquitylated RNAPII is cleared by the VCP segregase in the absence of TFIIH in the TCR complex. a** Top: Cartoon depicting the effect of UVSSA[KO] on TCR complex assembly. Bottom: Quantification of local DRB run-off assay as described for Fig. 1a-c RPE1 WT and UVSSA[KO] with or without VCPi treatment (NMS-873; 5 μM; 1 h pretreatment, then incubation for the indicated times after UV-C). **b** Representative images of RNAPII-S2P signal detected by western blot from whole cell lysates of primary fibroblasts from normal (48BR) or UVSSA-defective cells (Kps3) at 0, 2, 4, and 6 h after global UV-C (10 J/m²). Cells were incubated with VCPi (CB5083; 10 μM) for 1 h before UV-C and incubated in the same media. RNAPII-S2P was detected with an antibody against phosphorylated C-terminal domain (CTD) Ser2/5 (BioLegend, 920203 (H5)). Short and long exposures are shown (top and middle panels, respectively). SMC3 was used as loading control. This experiment was repeated three times. **c** Detection of RNAPII-S2P ubiquitylation (Cell Signaling, 3936) and TFIIH recruitment (p89 and p62) by western blot after immunoprecipitation of RNAPII-S2P (right) from chromatin

fractions (left) of RPE1 WT and indicated TCR KOs. Cells were incubated with or without VCPi (NMS-873; 5 μM) for 1 h before 12 J/m² UV-C. Then, cells were incubated in the same media and collected 1 h after UV-C. This experiment was repeated three times. **d** Detection of RNAPII-S2P ubiquitylation (Cell Signaling, 3936) and TFIIH recruitment (p89 and p62) by western blot after immunoprecipitation of RNAPII-S2P (right) from the chromatin fraction (left) of RPE1 WT cells or UVSSA[KO] with or without complementation of the indicated UVSSA mutants. Cells were treated with 12 J/m² UV-C and collected after 1 h. This experiment was repeated three times. **e** Top: Cartoon depicting the effect of UVSSA[ΔTFIIH] on TCR complex assembly. Bottom: Quantification of the local DRB run-off immunofluorescence assay as described for Fig. 1a-c in RPE1 WT, UVSSA[KO], and UVSSA[KO] stably expressing the indicated UVSSA constructs and with or without VCPi treatment as in (**a**). **f** Top: Cartoon depicting the effect of STK19[KO] on TCR complex assembly. Bottom: Quantification of the local DRB run-off immunofluorescence assay as described for Fig. 1a-c in RPE1 WT and STK19[KO] with or without VCPi treatment as in (**a**).

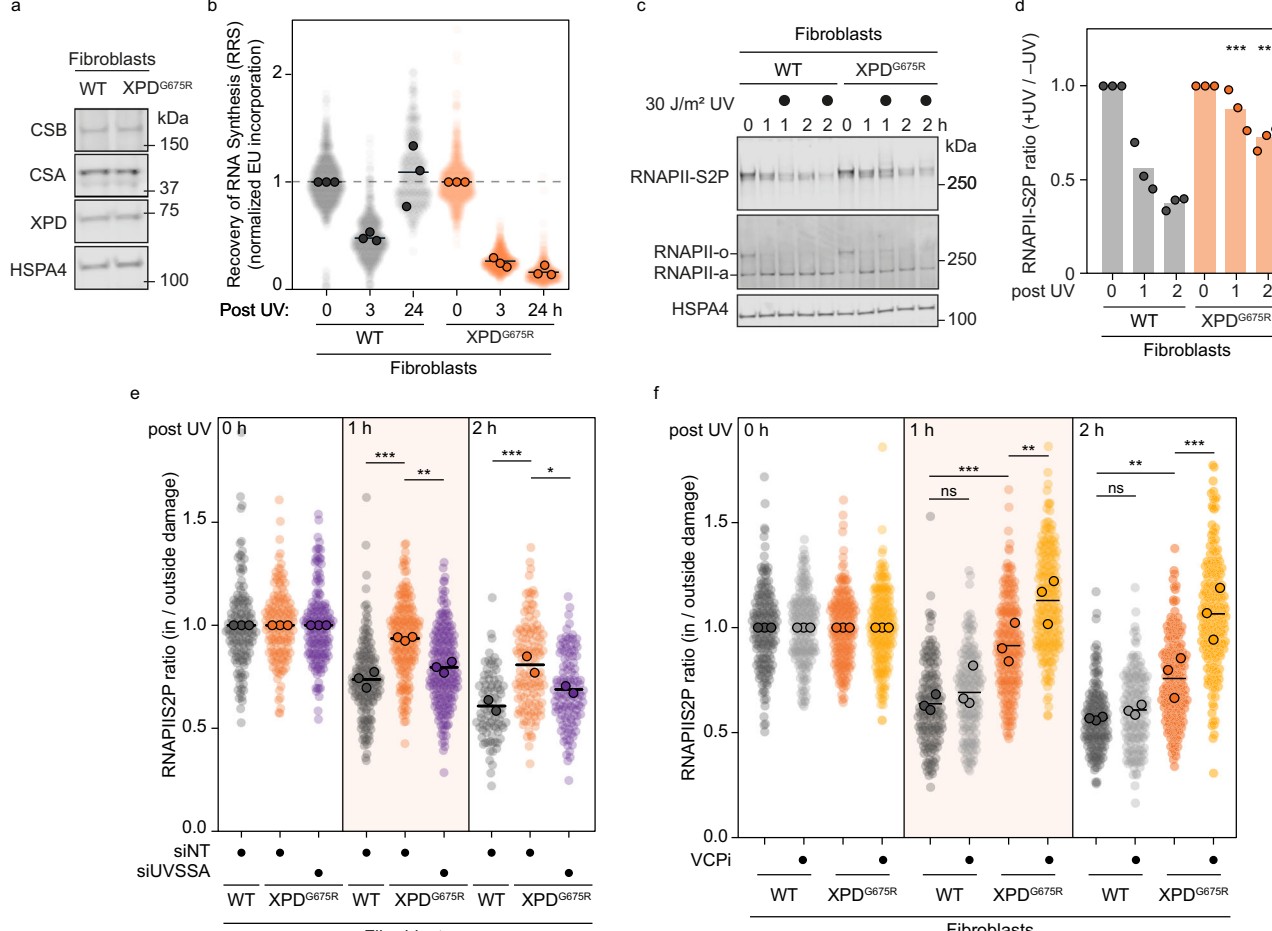

**Fig. 5 | Helicase-dead XPD impairs RNAPII clearance and necessitates VCP-dependent clearance in fibroblasts. a** Detection of CSB, CSA, and XPD protein levels in whole cell lysates from fibroblasts derived from an individual with a helicase-dead XPD mutant protein (XP8BR-hTERT, XPD[G675R]) or a normal individual (48BR-hTERT, WT). HSPA4 was used as a loading control. **b** Quantification of RRS as in Fig. 2a-b in normal and XPD helicase-dead fibroblasts. **c** A representative image of the global DRB run-off assay by western blotting in normal and XPD helicase-dead

fibroblasts as described for Fig. 1e. **d** Quantification of (**c**) as described for Fig. 1f-g. **e** Quantification of the local DRB run-off immunofluorescence assay as described for Fig. 1a-c for 0, 1, and 2 h after local UV-C treatment in normal and XPD helicase-dead fibroblasts after knockdown with an siRNA pool for UVSSA or a nontargeting control siRNA. **f**. Quantification of the local DRB run-off immunofluorescence assay as described for Fig. 1a-c for 0, 1, and 2 h after local UV-C treatment in normal and XPD helicase-dead fibroblasts with or without VCPi treatment as in Fig. 4a.

We next asked whether these findings were unique to primary fibroblasts or also observed in RPE1-hTERT cells. To test this, we stably expressed GFP-tagged XPD[WT] or one of two helicase-dead mutants (XPD[K48R] or XPD[G675R]) in RPE1-hTERT cells. The cells were then transfected with a synthetic guide RNA (crRNA) that targets only the endogenous XPD gene, leaving the integrated XPD cDNA intact

(Fig. 6a). This "switch" approach generates a complete endogenous XPD knockout while maintaining TFIIH stability through the helicase-dead mutants. Importantly, cells expressing only ectopic XPD[WT]-GFP were fully proficient in TCR. In contrast, expression of either helicase-dead XPD mutant caused a strong TCR deficiency, evidenced by the complete inability to recover RNA synthesis after UV irradiation

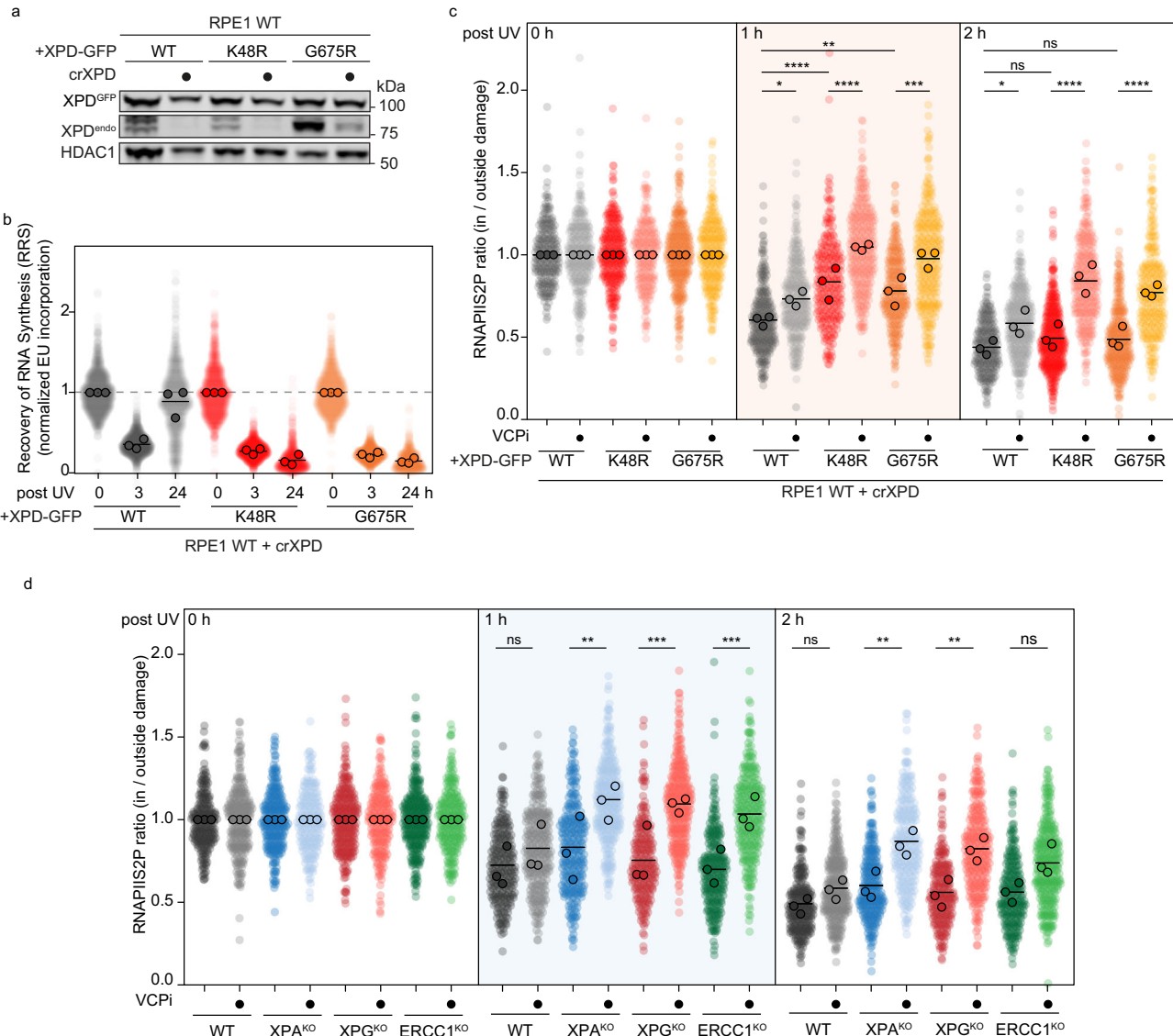

**Fig. 6 | XPD helicase activity drives RNAPII clearance during proficient TCR.**
**a** Detection of XPD levels by western blot in whole cell lysates of RPE WT cells stably expressing XPD^WT or one of two helicase-dead XPD mutants (XPD^K48R or XPD^G675R), with or without transfection of crXPD targeting only endogenous XPD. HDAC1 was used as a loading control. **b** Quantification of RRS as in Fig. 2a-b in RPE WT stably expressing the indicated XPD constructs after crXPD transfection. **c** Quantification

of the local DRB run-off immunofluorescence assay as described for Fig. 1a-c for 0, 1, and 2 h after local UV-C treatment in RPE WT stably expressing the indicated XPD constructs after crXPD transfection with or without VCPi treatment as in Fig. 4a. **d** Quantification of the local DRB run-off immunofluorescence assay as described for Fig. 1a-c for 0, 1, and 2 h after local UV-C treatment in RPE WT or the indicated TCR KO cells with or without VCPi treatment as in Fig. 4a.

(Fig. 6b). Consistent with the results in primary fibroblasts, RPE1-hTERT cells exhibited a pronounced RNAPII clearance delay under XPD helicase-dead conditions, with residual clearance becoming fully dependent on VCP (Fig. 6c). Together, these data demonstrate that XPD helicase activity drives normal RNAPII clearance during proficient TCR, and that in its absence, stalled RNAPII is extracted by VCP. Inhibition of both the TFIIH- and VCP-dependent pathways completely abolishes RNAPII clearance.

### Downstream NER factors stimulate XPD helicase to promote RNAPII clearance

So far, we have observed that impaired TFIIH recruitment due to UVSSA loss, mispositioning caused by STK19 loss, or impaired helicase activity from helicase-dead XPD leads to delayed RNAPII removal, which is partially compensated by slower VCP-mediated extraction. TFIIH is recruited to RNAPII as a core complex of seven subunits bound to the CAK module, consisting of three subunits[9,10], which inhibits XPD

helicase activity[50]. Activation of XPD helicase during GGR requires XPA, which triggers the release of the CAK module[51], and both XPA and XPG promote XPD helicase activity on naked DNA in vitro[36]. Superimposition of available cryo-EM structures suggests that ERCC1-XPF is positioned close to XPD[52], consistent with AlphaFold modeling, which also indicates a potential interaction between XPD and ERCC1-XPF (Supplementary Fig. 6b-c). We therefore asked whether stimulation of XPD by downstream NER factors contributes to TFIIH-driven RNAPII clearance during TCR. Measuring RNAPII removal in the presence of VCP inhibition, we found that clearance in XPA^KO, XPG^KO, and ERCC1^KO cells became fully dependent on VCP (Fig. 6d). These results suggest that efficient TFIIH-driven clearance of damage-stalled RNAPII during functional TCR may be facilitated by stimulation from downstream NER proteins, whereas in their absence, RNAPII removal appears to rely predominantly on VCP-mediated mechanisms. Although XPA, XPG, and ERCC1-XPF have not previously been detected in the same complex as damage-stalled RNAPII[9], our findings imply that these

downstream NER factors are recruited to TFIIH while TFIIH is bound to RNAPII, presumably to promote XPD helicase activity. A possible explanation for why current approaches have not detected this interaction is that, upon NER factor binding, TFIIH rapidly adopts its active repair conformation, and RNAPII is cleared, resulting in a very short-lived interaction that is difficult to capture. Future studies should aim to capture this interaction more accurately and assess its effects on TFIIH helicase activity and RNAPII clearance, for example, using proximity-labeling and biochemical approaches.

## Discussion

Stalling of elongating RNAPII at DNA lesions initiates the sequential recruitment of TCR factors, followed by nucleotide excision repair (NER) to remove the lesion. What happens to RNAPII itself during this process, however, has remained unclear. In this study, we established a time-resolved imaging approach to track the clearance of RNAPII at UV-induced lesions, combined with assays for its ubiquitylation and degradation in an isogenic TCR knockout collection. This allowed us to directly compare how individual TCR factors influence RNAPII processing. We show that CSB and CSA are essential for RNAPII ubiquitylation, clearance, and degradation, underscoring ubiquitylation as the key initiating step. Downstream of ubiquitylation, we uncovered two distinct clearance pathways. The primary pathway operates when TCR is intact: CSB, CSA, ELOF1, UVSSA, and STK19 promote RNAPII ubiquitylation and recruit and position TFIIH, whose XPD helicase, stimulated by XPA, XPG, and ERCC1-XPF, ultimately drives RNAPII removal, followed by NER-mediated excision of the lesion. A secondary pathway comes into play when TFIIH function is impaired, in which the ubiquitin-dependent VCP/p97 segregase extracts ubiquitylated RNAPII independently of DNA repair. Both pathways require CSB/CRL4$^{CSA}$-mediated ubiquitylation, explaining why congenital loss of CSB or CSA causes persistent RNAPII clearance defects (Fig. 7).

### A new method to monitor RNAPII clearance

The precise regulation and fate of damage-stalled RNAPII during TCR have remained unclear, largely due to the lack of experimental tools to address this directly[1,6,30,31]. Two recent studies introduced methods to measure RNAPII removal from damaged DNA, either by microscopy-based fluorescence recovery after photobleaching (FRAP) in live cells[32] or by a sequencing-based approach termed PADD-seq[33]. While highly informative, both methods require engineered cell lines, labor-intensive processing, and costly high-end imaging or sequencing. To overcome these limitations, we developed the local and global DRB run-off assays, which enable systematic and efficient analysis of the mechanisms governing RNAPII clearance. These assays are broadly applicable in many laboratory settings. Beyond enabling side-by-side comparison of isogenic TCR knockouts, the local DRB run-off assay can be applied to any adherent cell type amenable to microscopy. Moreover, it may be used to study RNAPII clearance in response to other types of DNA damage, provided local DNA damage can be induced. We envision particular applications in analyzing transcriptional responses to DNA double-strand breaks, where established methods for generating local lesions are already available.

### CSB and CSA are essential for lesion-stalled RNAPII clearance

To directly compare the role of individual TCR factors in the clearance and degradation of lesion-stalled RNAPII, we generated and validated a collection of eight isogenic knockout cell lines. All TCR-deficient lines were impaired in transcription recovery after UV irradiation and failed to generate trabectedin-induced DSBs, confirming their TCR deficiency. Using these eight knockouts in our local and global DRB run-off assays, we categorized them into three phenotypic groups based on their ability to clear lesion-stalled RNAPII: (i) Defective clearance, (ii) Delayed clearance (to varying degrees), which is further exacerbated by VCP inhibition, and (iii) Normal clearance. Importantly, only CSB$^{KO}$

and CSA$^{KO}$ cells, but none of the other knockouts, showed a persistent defect in clearing RNAPII from UV damage sites. These findings suggest that prolonged RNAPII stalling and transcription arrest, rather than defective DNA repair, may underlie the neurodegenerative symptoms of Cockayne syndrome caused by congenital deficiencies in CSB or CSA, consistent with recent results using complementary approaches[32,33].

### Ubiquitylation as the universal trigger for RNAPII clearance

CSB and the CRL4$^{CSA}$ E3 ligase complex play an essential role in the ubiquitylation of lesion-stalled RNAPII and in the subsequent recruitment of downstream NER factors[10,14]. The downstream clearance pathways identified here are both initiated by RNAPII ubiquitylation: (i) TFIIH recruitment, which is strongly dependent on RNAPII ubiquitylation, with RPB1$^{K1268R}$ knock-in cells showing very low levels of TFIIH binding, and (ii) the ubiquitin-selective segregase VCP, which requires at least low levels of RNAPII ubiquitylation to extract the polymerase from chromatin. Ubiquitylation of damage-stalled RNAPII involves both proteolytic K48 and non-proteolytic K63 linkages, which are CSA-dependent[10] and stimulated by ELOF1 and UVSSA[14]. Although the RPB1$^{K1268}$ residue is most important during functional TCR, other residues or ubiquitin conjugated to additional TCR proteins may also contribute to VCP-dependent clearance[33]. Stalled RNAPII may bear diverse ubiquitin linkages, leading to distinct functional outcomes. Developing new tools to detect and manipulate these modifications will be essential to dissect their roles in determining RNAPII fate[53]. Interestingly, clearance of RNAPII in UVSSA$^{\Delta TFIIH}$ cells was initially VCP-dependent but later showed slow, VCP/TFIIH-independent removal, consistent with direct proteasomal degradation of chromatin-bound RNAPII when extraction is inefficient but ubiquitylation is normal (Supplementary Fig. 5c).

### VCP removes suboptimal TCR complexes

Intriguingly, cells with attenuated clearance (ELOF1$^{KO}$, UVSSA$^{KO}$, CSA$^{\Delta UVSSA}$, and UVSSA-defective fibroblasts) exhibit reduced, suboptimal levels of RNAPII ubiquitylation, which correlate with slower RNAPII clearance. A "last resort" pathway was previously proposed to clear RNAPII in UVSSA-deficient cells[54], with VCP implicated in this process[32,33]. However, the timing and mechanism of VCP action remained unclear. Our clearance assays using VCP inhibition demonstrated that removal of misassembled, under-ubiquitylated TCR complexes is fully dependent on VCP in cell lines with attenuated clearance. Importantly, analysis of cells with normal RNAPII ubiquitylation but impaired TFIIH recruitment, positioning, or helicase activity (e.g., UVSSA$^{\Delta TFIIH}$, STK19$^{KO}$, XPD$^{K48R}$, XPD$^{G675R}$) revealed a similar delay in RNAPII removal, compensated by VCP-dependent extraction. These findings suggest that impaired TFIIH recruitment and its function, not reduced RNAPII ubiquitylation, is the main trigger for VCP-mediated clearance. Nevertheless, VCP requires a minimal level of RNAPII ubiquitylation, which is present in all tested lines except CSB$^{KO}$ and CSA$^{KO}$, explaining why both TFIIH- and VCP-dependent pathways fail in these cells. The existence of the VCP-driven backup pathway highlights the critical importance of removing stalled RNAPII, even when lesion repair by TCR is not possible.

### XPD helicase is the primary driver of RNAPII removal

Experiments in primary fibroblasts and engineered XPD helicase-dead cell lines (XPD$^{K48R}$ and XPD$^{G675R}$) suggest that XPD's helicase function is the driving force behind the clearance of RNAPII when TCR is fully functional. Therefore, we hypothesize that VCP removes RNAPII only when TFIIH fails to do so. A mechanism of TFIIH-driven RNAPII removal could be the result of two opposing ATPase activities: (i) XPD is positioned on the downstream DNA in front of RNAPII and pulls the DNA towards it to reach the lesion, while (ii) CSB is bound to the upstream DNA and also pulls on DNA towards it, in the opposite direction

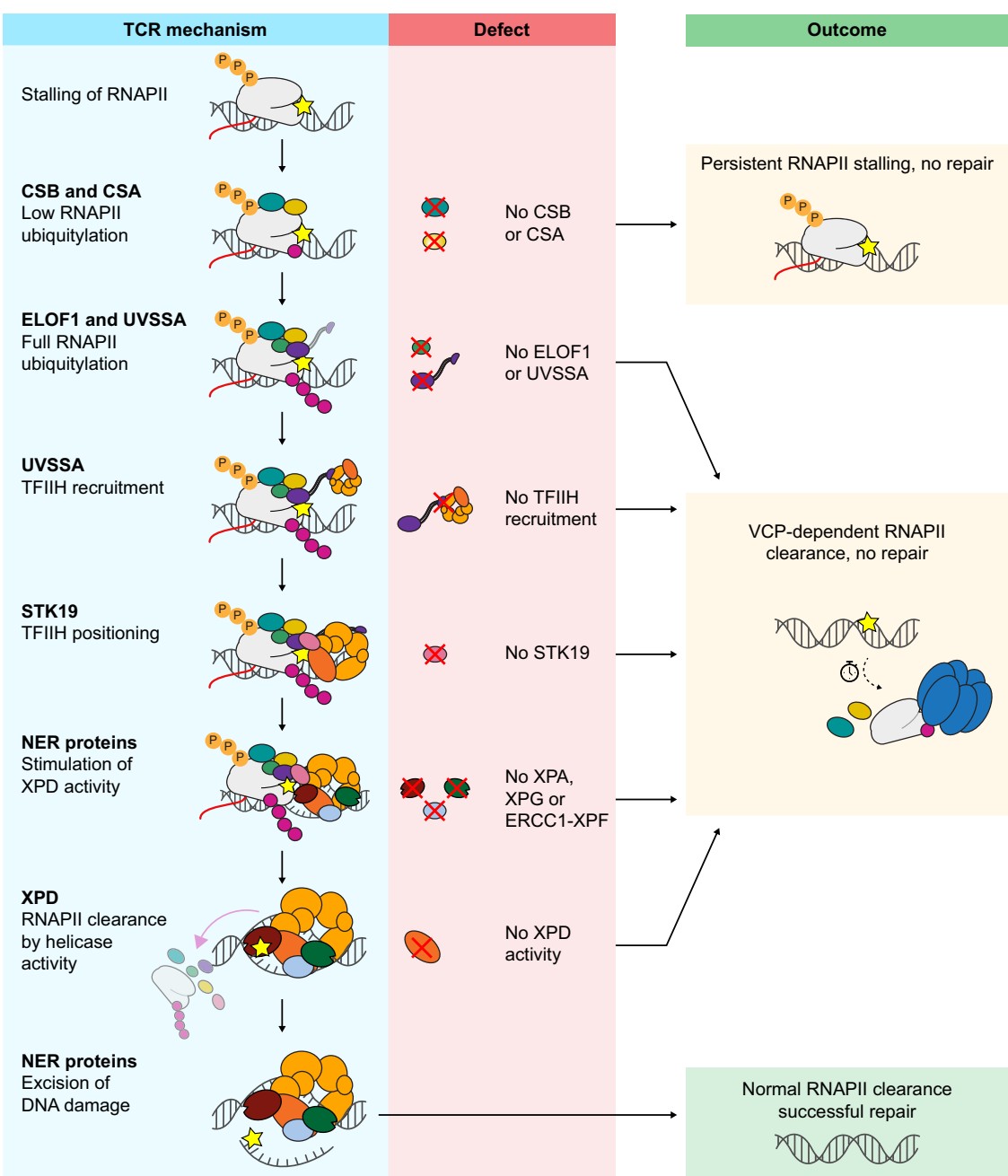

**Fig. 7 | Overview of the TCR mechanism, possible defects, and resulting outcomes.** Model depicting the TCR mechanism when fully functional in blue (left), defects that may occur (middle, red), and their resulting outcomes (right, orange/green). Damage-stalled RNAPII is only cleared by XPD helicase under conditions of proficient TFIIH recruitment, positioning, and activity stimulation, allowing repair

of the lesion. Any impairment of TFIIH function downstream of RNAPII ubiquitylation by CSB and CSA function leads to VCP-dependent clearance of damage-stalled RNAPII, without repair of the damage. Defects in CSB and CSA prevent any ubiquitylation of damage-stalled RNAPII, thus prohibiting both clearance pathways.

relative to XPD. The net results of these pulling forces could be the destabilization and dissociation of RNAPII, thereby enabling the next step in repair with TFIIH already bound to the lesion. Interestingly, we observe that helicase-dead XPD triggers a more pronounced RNAPII removal defect than not recruiting (UVSSA[KO]) or positioning (STK19[KO]) TFIIH altogether. Similarly, knockdown of UVSSA in primary fibroblasts with XPD[G675R] significantly improves polymerase removal. Thus, we speculate that VCP-mediated extraction is partially blocked once TFIIH associates with and is docked correctly onto the RNAPII-bound TCR complex. In situations where TFIIH is loosely associated with the polymerase through a flexible interaction with UVSSA but not fully engaged with the TCR complex, VCP can still have access to extract

RNAPII. However, once TFIIH is recruited, its proper positioning allows the timely removal of RNAPII. Superimposing an STK19-XPD AlphaFold model onto experimental cryo-EM structures of the RNAPII-bound TCR complex, including STK19, positions XPD very close to the RPB1[K1268] ubiquitylation site[5,19,20] (Supplementary Fig. 6d). Although speculative, this suggests that engaged XPD near K1268 could partially block VCP access for ubiquitin-dependent polymerase extraction.

Together, our results indicate a "shake it off, break it off model", wherein TFIIH is the primary route to dissociate ("shake off") the polymerase, enabling proper and timely TCR. However, in case this fails, VCP extracts the polymerase and shuttles it to the proteosome ("break it off"). Although currently out of reach, in vitro reconstitution

of TCR and single-molecule analysis of these events will be required to visualize these steps directly. A first cell-free TCR system was recently established[20], which is an important first step toward this goal.

## Methods

### Cell lines and cultures

The RPE1-hTERT cell lines were cultured at 37 °C in an atmosphere of 5% $CO_2$ in DMEM GlutaMAX (Thermo Fisher Scientific) supplemented with penicillin/streptomycin (Sigma), and 8% fetal bovine serum (FBS; Bodinco BV or Thermo Fischer Scientific (Gibco)). The patient-derived cell lines (Fig. 4d) were maintained in DMEM (WAKO) supplemented with 10% FCS (Hyclone) and antibiotics unless otherwise noted. All cell lines are listed in Supplementary Table 1.

### Generation of TCR knockout cells

Parental RPE1-hTERT cells stably expressing inducible Cas9 (iCas9) that are also knockout for TP53 and the puromycin-N-acetyltransferase PAC1 gene were described previously (referred here as RPE1 WT)[12]. RPE1 WT cells were transfected with Cas9-T2A-EGFP (pX458; Addgene #48138) containing a gRNA targeting CSB, CSA, ELOF1, UVSSA, STK19, XPA, XPG, or ERCC1 from the TKOv3 library using Lipofectamine 2000 (Invitrogen). The sgRNAs are listed in Supplementary Table 2, and plasmids in Supplementary Table 3. Cells were FACS sorted on EGFP and plated at low density, after which individual clones were isolated, expanded, and verified by western blot analysis and/or Sanger sequencing using the oligos listed in Supplementary Table 4.

### Validation of TCR knockout cells by Sanger sequencing

Genomic DNA was isolated by resuspending cell pellets in whole cell lysate (WCE) buffer (50 mM KCL, 10 mM Tris pH 8.0, 25 mM $MgCl_2$, 0.1 mg/mL gelatin, 0.45 % Tween-20, 0.45 % NP-40) containing 0.1 mg/mL Proteinase K (EO0491; Thermo Fisher Scientific) and incubating for 1 h at 56 °C followed by a 10 min heat inactivation of Proteinase K at 96 °C. Fragments of approximately 1 kb spanning the introduced mutations were PCR amplified, followed by Sanger sequencing using the primers listed in Supplementary Table 4.

### Recovery of RNA synthesis (RRS)

Cells were irradiated with UV-C light (9 J/m²) or not and incubated for the indicated periods, then pulse-labeled with 400 μM 5-ethynyl-uridine (EU; Jena Bioscience) for 1 h followed by a 15 min medium-chase with DMEM without supplements. Cells were fixed with 3.7% formaldehyde in phosphate-buffered saline solution (PBS) for 15 min, permeabilized with 0.5% Triton X-100 in PBS for 10 min at room temperature, and blocked in 1.5% bovine serum albumin (BSA, Thermo Fisher) in PBS. Nascent RNA was visualized by Click-iT chemistry, labeling the cells for 1 h with a mix of 60 μM Atto azide-Alexa594 (Atto Tec), 4 mM copper sulfate (Sigma), 10 mM ascorbic acid (Sigma), and 0.1 μg/mL DAPI in a 50 mM Tris-buffer (pH 8). Cells were washed extensively with PBS and mounted in Polymount (Brunschwig).

### Incision assay (γH2AX after trabectedin)

Cells were treated with 10 nM trabectedin (MedChemExpress) for 4 h. During the last 15 min, 20 μM 5-ethynyl-2′-deoxyuridine (5-EdU; Jena Bioscience) was added. Cells were then fixed with 3.7% formaldehyde in PBS for 15 min at room temperature. Cells were subsequently permeabilized with 0.5% Triton X-100 in PBS for 10 min at room temperature and blocked in 3% bovine serum albumin (BSA, Thermo Fisher) in PBS. Dividing cells were visualized by Click-iT chemistry, labeling the cells for 30 minutes with a mix of 6 μM Atto azide-Alexa594 or Atto azide-Alexa647 (Atto Tec), 4 mM copper sulfate (Sigma), and 10 mM ascorbic acid (Sigma) in a 50 mM Tris-buffer (pH 8). After washing with PBS, cells were blocked with 100 mM glycine (Sigma) in PBS for 10 min at room temperature and subsequently with 0.5% BSA and 0,05% Tween 20 in PBS for 10 minutes at room temperature. To

visualize γH2AX, cells were incubated with a primary antibody for phospho-Histone H2A.X Ser139 (JBW301, Merck) for 2 h at room temperature, followed by a secondary antibody Anti-Mouse Alexa 555(A-21424, Thermo Fisher) or Anti-Mouse Alexa 647 (A-21235), and 0.1 μg/mL DAPI for 1 h at room temperature and mounted in Polymount (Brunschwig).

### Immunoprecipitation of RNAPII-S2P

Cells were either mock-treated or irradiated with UV-C light (9 J/m²) and harvested 1 h after UV-C exposure. VCPi (NMS-873; Selleckchem, No. S7285) was added to the cells at a final concentration of 5 μM 1 h before UV-C irradiation, and then harvested 1 h after UV-C exposure. For endogenous RNAPII-S2P IPs, chromatin-enriched fractions were prepared by lysing the cells for 20 min on a rotating wheel at 4 °C in 1 mL EBC-1 buffer (50 mM Tris [pH 7.5], 150 mM NaCl, 2 mM $MgCl_2$, 0.5% NP-40, and protease inhibitor cocktail (Roche)), followed by centrifugation, and removal of the supernatant. The chromatin-enriched cell pellets were resuspended in 1 mL ECB-1 buffer supplemented with 500 U/mL Benzonase® Nuclease (Novagen) and 2 μg RNAPII-S2P (ab5095, Abcam) for 1 h at 4 °C. Then, the salt concentration was increased to 300 mM NaCl, and the samples were incubated for another 30 minutes on a rotating wheel at 4 °C. Samples were then centrifuged for 10 minutes at 20.000 RCF at 4 °C. 50 μL of the supernatant was saved as the input fraction, and the rest was transferred to fresh tubes. The protein complexes were immunoprecipitated by incubation with 20 μL prewashed Protein A agarose beads (Millipore) for 90 minutes at 4 °C. After incubation, the beads were washed 6 times with ECB-2(300) buffer (50 mM Tris [pH 7.5], 300 mM NaCl, 1 mM EDTA, 0.5% NP-40, and protease inhibitor cocktail (Roche)). The samples were then analyzed by western blotting.

### Western blotting

Total cell lysates were harvested by scraping the cells into Laemmli-SDS sample buffer. Total cell lysates, chromatin input, and IP samples were boiled for 10 min at 95 °C. Proteins were separated on Criterion™ XT Tris-Acetate 3−8% Protein Gels (BioRad, #3450131) in Tris/Tricine/SDS Running Buffer (BioRad, #1610744) or on Criterion Xt bis-tris 4-12% gels in MOPS running buffer. PVDF membranes (IPFL00010, EMD Millipore) were activated for 30 seconds in 100% methanol. Then, gels were blotted onto PVDF membranes in Tris/glycine blotting buffer (0.025 M Tris, 0.192 M glycine) with 20% methanol. Membranes were blocked with 5% fat-free milk in PBS with 0.1 % Tween-20 for 1 h at room temperature. Membranes were then probed with indicated antibodies in 5% fat-free milk in PBS with 0.1 % Tween-20 (Antibodies are listed in Supplementary Table 5). Proteins were stained with fluorochrome-conjugated secondary antibodies and were detected on an Odyssey CLx system and Image Studio software (Li-Cor).

### Local DRB-run-off immunofluorescence microscopy assay

Cells were plated in DMEM supplemented with 8% FBS, followed by serum starvation in DMEM without FBS for at least 24 h to reduce the number of replicating cells. Cells were locally UV-C irradiated through 5 μm pore filters (Milipore; TMTP04700) with 100 J/m² and immediately treated with 100 μM DRB (Sigma, D1916) for the indicated time periods, washed with PBS, and fixed with 3.7% formaldehyde in PBS for 15 min at room temperature. Cells were then permeabilized with 0.5% Triton X-100 in PBS for 10 min at room temperature. After washing with PBS, cells were blocked with 100 mM glycine (Sigma) in PBS for 10 min at room temperature and subsequently with 0.5% BSA and 0,05% Tween 20 in PBS for 10 min at room temperature. To visualize elongating RNAPII, cells were incubated with a primary antibody for RNAPII-S2P (ab5095, Abcam) and for CPD damages (CAC-NM-DND-001, Cosmo Bio) for 2 h at room temperature and then with secondary antibodies Anti-Mouse Alexa 488 (A-11029, Thermo Fisher) and Anti-Rabbit Alexa 555 (A-21429, Thermo Fisher) and 0.1 μg/mL DAPI for 1 h

at room temperature. Cells were subsequently mounted in Polymount (Brunschwig).

## Microscopic analysis of fixed cells

Images of fixed samples were acquired on a Zeiss AxioImager M2 widefield fluorescence microscope equipped with 63x PLAN APO (1.4 NA) oil-immersion objectives (Zeiss) and an HXP 120 metal-halide lamp used for excitation. Fluorescent probes and proteins were detected using the following filters for DAPI (excitation filter: 350/50 nm, dichroic mirror: 400 nm, emission filter: 460/50 nm), GFP (excitation filter: 470/40 nm, dichroic mirror: 495 nm, emission filter: 525/50 nm), Alexa 555/594 (excitation filter: 545/25 nm, dichroic mirror: 565 nm, emission filter: 605/70 nm), or Alexa 647 (excitation filter: 640/30 nm, dichroic mirror: 660 nm, emission filter: 690/50 nm). Images were recorded using ZEN 2012 software (Blue Edition, version 1.1.0.0) and analyzed in ImageJ (1.48 v).

## Global DRB run-off assay by western blotting

Cells were cultured in DMEM supplemented with 8% FBS. Then, cells were globally UV-C irradiated with 30 J/m$^2$ and immediately treated with 100 μM DRB (Sigma, D1916) for the indicated time periods. Cells were washed with PBS, and total cell lysates were harvested by scraping cells in the Laemmli-SDS sample buffer. Proteins were separated and analyzed by western blotting.

## Generating RPE1 cells complemented with CSA, UVSSA, and XPD variants

For lentiviral particle production, the GFP in the lentiviral vector pLenti-PGK-GFP-Puro was replaced with GFP-CSA$^{WT}$, GFP-CSA$^{\Delta CRL4}$ (D266A), GFP-CSA$^{\Delta UVSSA}$ (Y334A), XPD$^{WT}$-GFP, XPD$^{K48R}$-GFP, or XPD$^{G675R}$-GFP. Then, HEK293T cells (American Type Culture Collection, CRL-3216) were transfected with vectors expressing one of the GFP fusion proteins, VSV-G, RRE, and REV using PEI (Sigma-Aldrich). The virus-containing supernatant was collected after 48 h and filtered with a 0.44-μm filter. RPE1-hTERT CSA$^{KO}$ was transduced with lentiviral particles in the presence of 4 μg/mL Polybrene (Sigma-Aldrich) and 10 mM HEPES (pH 7.6). After 24 h, cells were selected with 1 μg/mL Puromycin. The generation of RPE1-hTERT UVSSA$^{KO}$ cells expressing GFP-UVSSA$^{WT}$ or GFP-UVSSA$^{\Delta TFIIH}$ (a Zn-finger mutant with the following substitutions: C567A-C577A-C585A-H588A) was described previously[14]. The expression of GFP-tagged CSA, UVSSA, and XPD proteins was verified using fluorescence microscopy and western blotting.

## Knockdown of genes using crRNA or siRNA

crRNA was used to knock down endogenous XPD in cells complemented with exogenous XPD variants. RPE1-hTERT cells stably expressing inducible Cas9 (iCas9) were incubated with 200 ng/mL Doxycycline for 24 h, then transfected with 10 nM tracRNA and 10 nM crRNA targeting endogenous but not exogenous XPD using RNAiMax Lipofectamine (Thermo Fisher Scientific, 13778150). 3 days after transfection, cells were collected and reseeded for the respective experiments. Knockdown of endogenous XPD was verified by western blotting using an XPD antibody (ab54676). To knock down UVSSA, XP8BR cells expressing helicase-dead XPD$^{G675R}$, were transfected with 40 nM ON-TARGETplus Human UVSSA siRNA smartpool (L-024139-02-0005; Horizon Discovery/Dharmacon) using RNAiMax Lipofectamine (Thermo Fisher Scientific, 13778150). The crRNA and siRNAs are listed in Supplementary Table 2, and the antibodies are listed in Supplementary Table 5.

## Detection of the elongating RNAPIIo after UV irradiation in primary fibroblast cell lines

Cells were cultured in a medium supplemented with 100 μM cycloheximide and 10 μM VCPi (CB5083, Selleck) for 1 h before UV-C irradiation. Cells were irradiated (10 J/m$^2$ of UV-C) and incubated for the indicated time periods in a medium containing cycloheximide and VCPi. Whole-cell lysates were resolved on 6% SDS-PAGE gels and transferred to the PVDF membrane. RNAPIIo was detected with an 3E10 or H5 antibody (recognizing phosphorylated CTD-Ser2), as indicated in the figure legends.

## Statistical analyses

In dot plots of immunofluorescence data, all values of cells are depicted as individual semi-transparent data points. The black horizontal line represents the mean of all data points. For bar graphs, the bar represents the mean of three independent biological replicates. All experiments in this study were executed independently three times ($n = 3$). The individual means of the three biological replicates are depicted as solid circles with black strokes. Statistical significance was tested with the individual means of the three biological replicates through an ordinary one-way ANOVA followed by Šídák's multiple comparisons test (two-tailed) using GraphPad Prism version 10.2.3 (403). ns: $p > 0.05$; *: $p < 0.05$; **: $p < 0.01$; ***: $p < 0.001$; ****: $p < 0.0001$.

## Reporting summary

Further information on research design is available in the Nature Portfolio Reporting Summary linked to this article.

# Data availability

All data supporting the findings of this study are available within the paper and its Supplementary Information. Source data are provided with this paper.

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

## Acknowledgements

We thank Annelotte P. Wondergem for generating XPA^(KO) cells. MSL laboratory was supported by the Netherlands Scientific Organization (ENW grant OCENW.M20.056, and VIDI grant ALW.016.161.320) and the European Research Council Consolidator Grant STOP-FIX-GO (grant agreement No 101043815). The TO laboratory was supported by the Practical Research Project for Rare / Intractable Disease (JP24ek0109765 to YN; JP23ek0109678 to TO), grants in Aid for Scientific Research KAKENHI from the Japan Society for the Promotion of Science (JP24K02223 to YN; JP23H00516 to TO), and the JST FOREST Program (JPMJFR221E to YN). The funders had no role in study design, data collection and analysis, the decision to publish, or the preparation of the manuscript.

## Author contributions

PJvdM performed all the local DRB run-off assays, incision assays, and detection of Ub-RNAPIIo in RPE1 cells. GY performed all the global DRB run-off and RRS assays, validated the isogenic collection of TCR knockout cell lines, reconstituted and validated CSA, UVSSA, and XPD mutants, and performed RNAPII-S2P immunoprecipitations. KT and YN performed all the Ub-RNAPIIo detection in patient-derived cell lines. TO supervised KT. PJvdM, GY, TO, and MSL interpreted the data and wrote the manuscript. PJvdM created all cartoon images as original, custom artwork.

## Competing interests

The authors declare no competing interests.
