## [Transparent Peer Review file · Nature Communications]

Hierarchical mechanisms control the clearance of DNA lesion–stalled RNA polymerase II

Corresponding Author: Professor Martijn Luijsterburg

Version 0:

Reviewer comments:

Reviewer #1

(Remarks to the Author)

The manuscript by Van der Meer et al., entitled, “Clearance of DNA damage-arrested RNAPII is selectively impaired in Cockayne syndrome cells,” investigated to what extent knockout or mutation of different transcription coupled-nucleotide excision repair (TC-NER) factors in human cells affect clearance of RNA Polymerase II (Pol II) stalled at UV-induced DNA lesions. To this end, they develop a new microscopy-based assay, which they call ‘local DRB run-off’, to examine clearance of hyper-phosphorylated elongating Pol II in response to localized DNA damage (in this case, localized UV irradiation). Importantly, DRB treatment prevents subsequent initiation of Pol II elongation, allowing the authors to specifically track the fate of already elongating Pol II. They construct and analyze knockouts in seven different TC-NER genes (CSA, CSB, ELOF1, UVSSA, XPA, XPF, and XPG) in immortalized RPE cells. While all of these mutants affect TC-NER (based on their Trabectedin/gammaH2AX assay) and ATF3 protein levels, their data indicate that only CSB/CSA cause a complete defect in clearance of hyper-phosphorylated Pol II following localized UV radiation. Similar results were obtained through their global DRB run-off assay. Based on these and other data, the authors suggest that the reason mutations in CSA and CSB genes cause the more severe developmental and neurological symptoms observed in Cockayne syndrome is due to the observed defects in clearance of stalled Pol II from DNA lesions, not defects in TC-NER. Overall, the experiments and data presented seem to be well done and informative, and the conclusions of the paper are interesting and significant. The following concerns should be addressed prior to publication.

Major concerns:

1. At multiple points in the manuscript, the authors’ draw the conclusion that their data indicates that CSA and CSB is required for eviction and degradation RPB1/RPB1 stalled at UV damage, e.g., “indicative of active degradation upon eviction from DNA lesions” (page 5), “only CSB and CSA knockout cells have defects in UV-induced ubiquitylation of stalled RNAPII, its clearance from damage sites, and degradation” (page 9), etc. To verify this conclusion, it would be helpful to include experiments using a proteasome inhibitor, in order to confirm that the loss of hyper-phosphorylated Pol II indeed reflects proteasome-mediated degradation.
2. Is it possible that dephosphorylation of the Pol II/RPB1 CTD by the FCP1/CTDP1 phosphatase contributes to loss of hyper-phosphorylated Pol II following UV irradiation? Do the authors’ data rule out this possibility?

Minor points:

3. Page 3: “individuals with the majority of UV-sensitive Syndrome (UVSS)...” – not clear what this means.
4. Page 6: “one local UV damage per nucleus using microfilters...” – maybe one ‘region’ or ‘focus’ of UV damage...?

Reviewer #2

(Remarks to the Author)

This study by van der Meer, Yakoub, et al, aims to shed light on the severity of defects observed in Cockayne Syndrome patients caused by mutations in CSB and CSA genes, compared to other TCR (Transcription Coupled Nucleotide Excision Repair) genes. They hypothesize that possible differences in the processing and degradation of the damage-arrested RNAPII may represent the molecular basis of the phenotypic variation observed between these patients.

The authors use, mostly, two complementary assays to (a) track the loss of elongating RNAPII in cells that have been irradiated locally by UV and (b) examine the levels of RNAPII after UV damage, in normal and (several) TCR-deficient cell lines. Specifically, they find that CSA and CSB knockout (KO) cells are characterized by a defect in the removal and degradation of RNAPII after UV, compared to the other isogenic TC-NER deficient cell lines.

Although the main finding of the paper is interesting for the TC-NER field and is supposed to examine the differences in the processing and degradation of the damage-arrested RNAPII more systematically, in essence, it confirms previous observations by Bregman et al, 1996 PNAS, Nakazawa et al, 2012 Nature Genetics and Nakazawa et al., 2020 Cell. Notably, another study published a year ago as a preprint (Hansen et al, 2023 Biorxiv) has examined the same question through a different methodology and reached similar conclusions. Since the whole manuscript is built around this, I believe that the study of van der Meer, Yakoub, et al, lacks the novelty required for being considered for publication in Nature Communications.

Below I state other major and minor comments, which I believe that by addressing them, the authors can improve this manuscript for publication elsewhere.

Major comments:

- In the Western Blots presented in Figure 1e (and others of similar experimental setup) the authors use whole cell extracts (WCE) to examine the rate of clearance of RNAPII from sites of damaged chromatin. Since they are interested in examining a phenomenon that occurs only in the context of the damage-arrested and chromatin-bound RNAPII, it is necessary to use the chromatin fraction (using the same setup) instead of whole cell lysates.
- A large part of the study deals with the characterization of the newly constructed TCR defective isogenic KO cell lines. The most classical way to approach this is to examine the ability of the cells to recover RNA synthesis (RRS) after exposure to UV damage. This important experiment is lacking. Instead, the authors present only indirect observations (e.g., they use as a proxy of TCR deficiency ATF levels in whole cell lysates 24 hours after exposure to UV. They also use trabectedin, which has reduced cytotoxic action in TCR-deficient cell lines; however, additional DNA damage response mechanisms have been implicated in its action, and thus it can only be used as a complementary assay to RRS to characterize TCR deficiency).

Minor comments:

- What is the reason that the authors use TP53 knock-out cells to construct their isogenic cell lines? As this will also compromise the efficiency of the second NER pathway (GGR) and render the cell more sensitive.
- Figure 1c, the y-axis label has a spelling mistake ("damge" instead of "damage").
- Figure 1e and other similar figures in the text. An indication that dots in the western blots represent the UV-treated samples is missing, so it is clearer for the reader.
- Figure 2a. RNAPII indication is missing from the schematic.

Reviewer #3

(Remarks to the Author)

The present study by van der Meer and Collaborators provide an interesting observation: cells knock out for either CSA or CSB genes display a failure in RNA polymerase II (RNAPII) clearance at the site of UV-induced DNA lesions, likely associated with a lack of RNAPII post-translational modifications. Mutations in either CSA or CSB genes are causative of Cockayne syndrome (CS), the severe Cerebro-oculo-facio-skeletal syndrome (COFS) or the very mild UV sensitive syndrome (UVSS). Failure of RNAPII clearance is not observed in cells defective for other genes encoding proteins involved in DNA repair by nucleotide excision repair (NER) and associated to another pathological condition, the xeroderma pigmentosum. Based on these findings, the Authors suggest that RNAPII clearing defects may underlie the Cockayne syndrome-like neurodegenerative phenotype.

While the observations may be interesting and worth of further investigations, the findings are still too preliminary and the conclusions not supported by the patient's clinical spectrum. Overall, the data cannot justify a publication in Nature Communications.

Major concerns:

- Defects in CSB or CSA genes are not always associated to neurological defects and/or neurodegeneration. In particular, Horibata K and Colleagues (doi: 10.1073/pnas.0404587101) demonstrated that homozygous mutation in the human genome leading to the loss of CSB protein synthesis (a nonsense mutation in position 77 of CSB) resulted in a UVSS patient with no neurological abnormalities;
- Nardo T and Colleagues (doi: 10.1073/pnas.0902113106) demonstrated that a missense mutation in position 361 of CSA resulted in failure of transcription-coupled repair (TCR) but not neurological abnormalities or neurodegeneration. These observations are in contrast with the Authors' claim that their findings can explain the Cockayne syndrome-like neurodegenerative phenotype. Several articles demonstrated that an accumulation of oxidative stress in CS cells, likely

caused by additional roles of CSA and CSB outside TCR, is more directly correlated with CS neurodegeneration.

-CS patients may display neurological defects and/or neurodegeneration. XPA-defective individuals do not display CS-like features, nevertheless they always exhibit neurodegeneration. Considering that XPA knock out cells generated in this study do not show RNAPII clearance defects, it is unlikely that “deficiency in RNAPII processing and prolonged transcription arrests in response to DNA damage, rather than compromised DNA repair, may underlie the Cockayne syndrome-like neurodegenerative phenotype”.

-The Authors claim that the appearance of the RNAPII-o upper band upon UV irradiation corresponds to the ubiquitinated form of elongating RNAPII (Fig. 5 and 6). Even though this may be likely, the statement should be proved with specific assays (using anti-ubiquitin antibodies, proteasome inhibitors, specific immunoprecipitations, etc.).

-While CSB, CSA, UVSSA can be defined TCR factors, XPA, XPG, ERCC1, TFIIH should be named NER factors (as their inactivation impairs both global genome and transcription-coupled repair).

-Statistical analyses are missing. Proper experimental controls are missing (negative controls such as cells without DRB treatment). No clear explanation is provided concerning the different type of RNAPII antibodies used (which population of phosphorylated forms of RNAPII are recognized).

Reviewer #4

(Remarks to the Author)

I co-reviewed this manuscript with one of the reviewers who provided the listed reports as part of the Nature Communications initiative to facilitate training in peer review and appropriate recognition for co-reviewers

Version 1:

Reviewer comments:

Reviewer #1

(Remarks to the Author)

The revisions to the manuscript have addressed my previous concerns and significantly improved the manuscript.

Reviewer #2

(Remarks to the Author)

Overall, I find this manuscript improved compared to the previous version.

However, the concern regarding novelty remains valid, as the study investigating the same mechanism and reaching similar conclusions was published in early September (Gonzalo-Hansen, et al., NAR) following its initial release as a preprint a year ago. Given that this manuscript is largely centred on this mechanism, I believe it lacks the novelty necessary for consideration for publication in Nature Communications.

Below are a few comments that need to be addressed.

Regarding my previous comment on chromatin fraction and RNAPII S2 detection, I agree with the authors that this elongating form of RNAPII is exclusively associated with chromatin. Since the authors use various antibodies to detect RNAPII phosphorylation status—some of which may lack precise specificity for RNAPII S2 versus S5 phosphorylation—they should indicate in the text or figure legends which antibody is used for each specific experiment. While Supplementary Table 5 lists the different RNAPII antibodies used for western blotting, the authors do not specify in the text which antibodies were applied in each experiment.

A couple of points that I unfortunately missed in the previous revision are the following:

In Figure 1e, the authors report that ‘The antibody against the N-terminal domain of RNAPII detects both initiating (RNAPIIa) and the elongating (RNAPIIo) forms (middle blot). Note that RNAPII-S2 and RNAPIIo represent the same pool of phosphorylated RNAPII.’

Could the authors clarify why RNAPII-S5 is not detectable in their extracts when using an antibody that targets the N-terminus of the RNAPII RPB1 subunit, which should theoretically recognize both non-phosphorylated and all the phosphorylated states of RNAPII? Please correct the text accordingly.

In a related context, the authors state that ‘DRB prevents new RNAPII complexes from transitioning from the initiation form (RNAPIIa) to the hyperphosphorylated elongation form (RNAPIIo/S2)’. Do the authors believe that DRB inhibits the transition from RNAPIIa to RNAPII-S2? This is established knowledge and should be accurately described and represented. Please revise the diagrams in Supplementary Figure 2 to reflect this accordingly.

Reviewer #3

(Remarks to the Author)

In the revised version of the manuscript "Clearance of DNA damage-arrested RNAPII is selectively impaired in Cockayne syndrome cells", Authors address some of the criticisms raised by the present Reviewer. By doing so, they reveal weakness that cannot be ignored and undermine the main conclusion: "a selective RNAPII clearance defect in CSA- or CSB-deficient cells, in contrast to loss of other TCR genes".

In detail:

1_ Authors claim, both in the Abstract and Introduction, that "only congenital defects in the CSA or CSB genes cause the neurodegenerative disorder Cockayne syndrome, which is not observed with other TCR genes, despite their equal importance in TCR"

Mutations in XPD, XPG, XPF or ERCC1 gene have shown to result in CS clinical features associated with XP (and belonging to different XP complementation groups). Despite what attested by the Authors, detailed clinical studies reveal that the neurodegenerative features in XP-CS patients fully overlap those of CS (such as hearing loss, loss of cognitive and motor skills, tigroid demyelination). Please, see: <https://ojrd.biomedcentral.com/articles/10.1186/s13023-017-0616-2>

In agreement with this observation, the Authors, find that the XP8BR-hTERT cells (from a XP-CS patient belonging to the XP-D complementation group) present a significant RNAPII clearance defect comparable to that of CS-B cells. Why then the claim "Using this method on an isogenic collection of TCR knockout cells or patient-derived cells reveals a selective RNAPII clearance defect in CSA- or CSB-deficient cells, in contrast to loss of other TCR genes"?

2_ According to the Reviewer's suggestion, Authors also investigate cells from the few UVSS patients with mutations in CSA or CSB genes and not neurological features described. Cells from both cases showed defects in RNAPII clearance. The Authors argue that the UVSS patients similarly to the late-onset CS cases may eventually develop neurological symptoms. This Reviewer thinks that this assumption is speculative. As far as the French patient is concerned (UVSS1VI), she is currently over 30, she is a healthy mother with no sign of neurodegeneration so far. The attempt to explain a pathological feature, such as neurodegeneration, through a molecular dysfunction cannot ignore the clinical features of all patients.

3_ The Authors disagree that an accumulation of oxidative stress is found in CS cells (rebuttal letter) and they claim (manuscript) that "the sensitivity to oxidative DNA damage is not a distinguish features of CS cells". Several laboratories worldwide have found that CS cells are characterized by mitochondrial dysfunction, high level of ROS, accumulation of oxidative lesions and even hypersensitivity to oxidative base damage. Here only some examples:

doi: 10.1086/380399 (and reference therein);
DOI: 10.1016/j.dnarep.2005.06.017
doi: 10.1016/j.dnarep.2005.06.017
doi: 10.1111/j.1474-9726.2012.00815.x
doi: 10.1093/hmg/dds211
doi: 10.1073/pnas.1414135111.
doi: 10.1016/s1568-7864(02)00188-x
doi: 10.1073/pnas.0902113106

BER is the pathway that removes most of the oxidative DNA modifications in the genome. Therefore, the finding that BER defects affect cell survival to oxidative stress more than TCR defects (Olivieri et al.) is not surprising. The Authors' conclusion is an oversimplification.

4_ Calling proteins that work both in GGR and TC-NER as TCR factors creates confusion. They should be called NER factors, like the scientific community does.

5_ The Authors refer to a previously published paper to validate RNAPII ubiquitylation at K1268. This is not fully correct, since they should include a direct validation in their current experimental system.

6_ Other minor points:

Figure 4C and Figure S5D should be placed together since one represents the western blot and the other its quantification (4C).

Where is the image quantified in Figure 4D? In addition, the Authors' conclusion does not explain why there is no effect in wild type cells

After carefully reading the revised version of the manuscript and the point-by-point rebuttal, the present Reviewer is convinced that the finding of the manuscript and the data interpretation are not sufficient for the Readers of Nature Communications journal. Part of their finding, that TCR defects are associated to the loss of RNAPII clearance, were previously found in Bregman, D.B. et al. UV-induced ubiquitination of RNA polymerase II: a novel modification deficient in Cockayne syndrome cells. Proc Natl Acad Sci U S A 93,11586-90 (1996). The manuscript could benefit from a deeper comparative analysis with earlier RNAPII degradation studies. Authors could also add a dedicated section contrasting their findings with prior literature to highlight advancements or deviations.

Reviewer #4

(Remarks to the Author)

Reviewer #5

(Remarks to the Author)

see confidential comments to the editors

Version 2:

Reviewer comments:

Reviewer #1

(Remarks to the Author)

The revised manuscript by Van der Meer et al., entitled: "Hierarchical mechanisms control the clearance of DNA lesion-stalled RNA polymerase II" is significantly improved by the inclusion of new data showing that the TFIIH subunit XPD plays an important role in RNA polymerase II removal following stalling at UV damage. The authors now suggest a mechanism in which XPD plays a major role in removing lesion-stalled polymerase and VCP/p97 segregase plays a backup role. While I thought the original version of the manuscript was acceptable, the new version is that much better. I had only one minor suggestion:

1. On line 305/306, the text, "separate-of-function mutant", I think should be: "separation-of-function mutant".

Reviewer #2

(Remarks to the Author)

The current revised manuscript has addressed adequately my concerns.

Reviewer #6

(Remarks to the Author)

This manuscript investigates how RNA polymerase II (RNAPII) is cleared from sites of transcription-blocking DNA lesions and proposes a hierarchical model in which TFIIH-driven clearance depends on stimulation by downstream nucleotide excision repair (NER) factors, while, in their absence, VCP (p97) mediates RNAPII removal. The authors build on their previously described DRB run-off assay to monitor RNAPII clearance kinetics in cells deficient for various TCR and NER components.

The study is technically strong and addresses an important question in transcription-coupled DNA repair (TCR). It provides the first systematic analysis of the contribution of downstream NER factors (TFIIH, XPA, XPG, ERCC1-XPF) to RNAPII removal, extending beyond earlier reports that focused primarily on upstream TCR components (CSB, CSA, UVSSA) (NAR, 2024, <https://doi.org/10.1093/nar/gkae618>; Nature Communications, 2024, <https://doi.org/10.1038/s41467-024-51463-x>; NAR, 2023, <https://doi.org/10.1093/nar/gkad008>). The data are well organized and of high quality. However, while the work provides valuable insights into the Pol II clearance process, a few points need to be clarified.

Major point

1. The manuscript proposes that downstream NER factors (XPA, XPG, ERCC1-XPF) stimulate XPD helicase to promote RNAPII clearance, implying that clearance occurs after the recruitment of these factors. However, a previous study (van der Weegen et al., Nat Commun 2020, <https://doi.org/10.1038/s41467-020-15903-8>) reported that although CS proteins and TFIIH readily assembled with RNAPII after UV irradiation, downstream repair proteins such as XPA and ERCC1-XPF could not be detected by immunoprecipitation. The authors should discuss this apparent contradiction.
2. The manuscript describes XPD helicase as the primary driver of RNAPII removal, which may be a somewhat over-interpretation of current data. While XPD helicase activity is clearly important for NER and could contribute to local DNA unwinding or remodeling around stalled RNAPII, direct solid biochemical evidence that XPD actively drives RNAPII displacement in TCR is still lacking. The authors may wish to tune down the statement and acknowledge that this remains a working hypothesis to be further tested in future mechanistic studies.
3. The manuscript states that downstream NER factors stimulate XPD helicase to promote RNAPII clearance. However, the authors may wish to clarify this point, as the direct evidence supporting this mechanism appears limited. A more open and cautious wording would be advisable when describing this relationship.

4. Previous studies have reported the involvement of upstream TCR components (CSB, CSA, UVSSA) in RNAPII clearance (NAR, 2024, <https://doi.org/10.1093/nar/gkae618> ; Nature Communications, 2024, <https://doi.org/10.1038/s41467-024-51463-x> ; NAR, 2023, <https://doi.org/10.1093/nar/gkad008>). The present study appears to report related findings, and it may be helpful for the authors to discuss their results in the context of these previous reports to provide a more balanced and integrated perspective.

5. The manuscript first states that “To confirm that the loss of hyperphosphorylated RNAPII-S2P indeed reflects proteasome-mediated degradation, we included the proteasome inhibitor MG132 in our experiments. Both local and global DRB run-off methods showed that inhibiting the proteasome abrogated the clearance and degradation of damage-stalled RNAPII from chromatin (Figure 1h–k)” (page 5–6, lines 138–142). Later, however, it is noted that “Treatment of cells with VCPi led to a small but nonsignificant effect on RNAPII clearance in WT cells, indicating that VCP-mediated chromatin extraction does not play a significant role in TCR-proficient cells (Figure 4a, Figure S5a)” (page 8, lines 275–278). As VCP/p97 is generally considered an essential cofactor in proteasome-mediated degradation, the authors may wish to clarify how these observations could be reconciled, or whether they reflect distinct experimental conditions or mechanisms of RNAPII clearance.

6. The manuscript focuses on VCP/p97-dependent RNAPII clearance but does not examine how this process relates to proteasome-mediated degradation. Given that VCP/p97 typically acts upstream of the proteasome to extract ubiquitylated substrates from chromatin, it remains unclear whether RNAPII removed in this manner is ultimately degraded or recycled. Previous work (Nature Communications, 2024, <https://doi.org/10.1038/s41467-024-51463-x>) has directly linked proteasome activity to RNAPII degradation under UVSSA-deficient conditions. The authors may wish to discuss how their findings align or differ from these results to clarify the mechanistic relationship between VCP and the proteasome.

Minor point

1. On page 13, line 488, the text reads “(Figure S5d)” in the sentence “Superimposing an STK19-XPD AlphaFold model onto experimental cryo-EM structures of the RNAPII-bound TCR complex, including STK19, positions XPD very close to the RPB1K1268 ubiquitylation site (Figure S5d)”. This appears to be a typo — the correct reference should be “(Figure S6d)”.

2. The manuscript employs the UVSSA Δ TFIIH mutant but does not cite a reference describing its design or the deleted region. The authors should indicate the source or provide details of the deleted amino acid sequence to ensure clarity and reproducibility.

Reviewer #7

(Remarks to the Author)

Reviewer #1 (Remarks to the Author)

The manuscript by Van der Meer et al., entitled, "Clearance of DNA damage-arrested RNAPII is selectively impaired in Cockayne syndrome cells," investigated to what extent knockout or mutation of different transcription coupled-nucleotide excision repair (TC-NER) factors in human cells affect clearance of RNA Polymerase II (Pol II) stalled at UV-induced DNA lesions. To this end, they develop a new microscopy-based assay, which they call 'local DRB run-off', to examine clearance of hyper-phosphorylated elongating Pol II in response to localized DNA damage (in this case, localized UV irradiation). Importantly, DRB treatment prevents subsequent initiation of Pol II elongation, allowing the authors to specifically track the fate of already elongating Pol II. They construct and analyze knockouts in seven different TC-NER genes (CSA, CSB, ELOF1, UVSSA, XPA, XPF, and XPG) in immortalized RPE cells. While all of these mutants affect TC-NER (based on their Trabectedin/gammaH2AX assay) and ATF3 protein levels, their data indicate that only CSB/CSA cause a complete defect in clearance of hyper-phosphorylated Pol II following localized UV radiation. Similar results were obtained through their global DRB run-off assay. Based on these and other data, the authors suggest that the reason mutations in CSA and CSB genes cause the more severe developmental and neurological symptoms observed in Cockayne syndrome is due to the observed defects in clearance of stalled Pol II from DNA lesions, not defects in TC-NER. Overall, the experiments and data presented seem to be well done and informative, and the conclusions of the paper are interesting and significant. The following concerns should be addressed prior to publication.

Major concerns:

(1). At multiple points in the manuscript, the authors' draw the conclusion that their data indicates that CSA and CSB is required for eviction and degradation RPB1/RPB1 stalled at UV damage, e.g., "indicative of active degradation upon eviction from DNA lesions" (page 5), "only CSB and CSA knockout cells have defects in UV-induced ubiquitylation of stalled RNAPII, its clearance from damage sites, and degradation" (page 9), etc. To verify this conclusion, it would be helpful to include experiments using a proteasome inhibitor, in order to confirm that the loss of hyper-phosphorylated Pol II indeed reflects proteasome-mediated degradation.

We performed the suggested experiments and carried out RNAPII clearance assays (imaging and WB) in the presence of proteasome inhibitor MG132, which showed strongly reduced clearance (Fig 1h-k). This suggests that the loss of hyper-phosphorylated RNAPII indeed reflects proteasome-mediated degradation. In addition, we tested VCP inhibitor as well, which had not appreciable impact in wild-type cells. However, in cells showing low levels of RNAPII ubiquitylation (ELOF1^{KO} and UVSSA^{KO}), we detect that RNAPII clearance becomes fully dependent on VCP activity (Fig 4d).

(2). Is it possible that dephosphorylation of the Pol II/RPB1 CTD by the FCP1/CTDP1 phosphatase contributes to loss of hyper-phosphorylated Pol II following UV irradiation? Do the authors' data rule out this possibility?

We cannot fully rule out a contribution by dephosphorylation. However, strongly arguing against this is that: (1) CSA^{KO} or CSB^{KO} cells show no loss of hyper-phosphorylated RNAPII but, in principle, there could be dephosphorylation, (2) proteasome inhibitor and (3) neddylation inhibitor (shown in Fig S5a-c) prevents the loss of hyper-phosphorylated RNAPII to a large extent.

Minor points:

(a). Page 3: "individuals with the majority of UV-sensitive Syndrome (UVSS)..." – not clear what this means.

We rephrased to for clarity to: '*Conversely, individuals with UV-sensitive Syndrome (UV^SS), caused by defective UVSSA, only display mild photosensitivity*'.

(b). Page 6: "one local UV damage per nucleus using microfilters..." – maybe one 'region' or 'focus' of UV damage...?

We rephrased as suggested to: '*.. one region of local UV damage per nucleus using microfilters*'.

Reviewer #2 (Remarks to the Author)

This study by van der Meer, Yakoub, et al, aims to shed light on the severity of defects observed in Cockayne Syndrome patients caused by mutations in CSB and CSA genes, compared to other TCR (Transcription Coupled Nucleotide Excision Repair) genes. They hypothesize that possible differences in the processing and degradation of the damage-arrested RNAPII may represent the molecular basis of the phenotypic variation observed between these patients.

The authors use, mostly, two complementary assays to (a) track the loss of elongating RNAPII in cells that have been irradiated locally by UV and (b) examine the levels of RNAPII after UV damage, in normal and (several) TCR-deficient cell lines. Specifically, they find that CSA and CSB knockout (KO) cells are characterized by a defect in the removal and degradation of RNAPII after UV, compared to the other isogenic TC-NER deficient cell lines.

Although the main finding of the paper is interesting for the TC-NER field and is supposed to examine the differences in the processing and degradation of the damage-arrested RNAPII more systematically, in essence, it confirms previous observations by Bregman et al, 1996 PNAS, Nakazawa et al, 2012 Nature Genetics and Nakazawa et al., 2020 Cell. Notably, another study published a year ago as a preprint (Hansen et al, 2023 Biorxiv) has examined the same question through a different methodology and reached similar conclusions. Since the whole manuscript is built around this, I believe that the study of van der Meer, Yakoub, et al, lacks the novelty required for being considered for publication in Nature Communications.

Below I state other major and minor comments, which I believe that by addressing them, the authors can improve this manuscript for publication elsewhere.

Major comments:

(1) In the Western Blots presented in Figure 1e (and others of similar experimental setup) the authors use whole cell extracts (WCE) to examine the rate of clearance of RNAPII from sites of damaged chromatin. Since they are interested in examining a phenomenon that occurs only in the context of the damage-arrested and chromatin-bound RNAPII, it is necessary to use the chromatin fraction (using the same setup) instead of whole cell lysates.

We disagree that it is necessary to use the chromatin fraction. Please note that phosphorylation of Ser2 only takes place on chromatin and nowhere else. RNAPII dephosphorylation is required for its dissociation and recycling. Therefore, we are monitoring – *per definition* – the chromatin-bound form of RNAPII.

The chromatin fractionation with increasing salt below (stained for S2P, adapted from <https://pubmed.ncbi.nlm.nih.gov/36291070/>) shows that Ser2-RNAPII is not present in the supernatant, but only in the 600 nM NaCl fraction and the insoluble chromatin fraction.

To demonstrate this point experimentally (see figure below; *not included in the revised manuscript*), we performed and quantified an RNAPII degradation assay in WT cells using either a whole cell lysate or the chromatin fraction. There is no difference in outcome, as expected, since phosphorylation of Ser2 only takes place on chromatin and nowhere else. This shows experimentally that it is – *in fact* – not necessary to use the chromatin fraction and that performing this analysis on whole cell lysates is just as informative.

(2) A large part of the study deals with the characterization of the newly constructed TCR defective isogenic KO cell lines. The most classical way to approach this is to examine the ability of the cells to recover RNA synthesis (RRS) after exposure to UV damage. This important experiment is lacking. Instead, the authors present only indirect observations (e.g., they use as a proxy of TCR deficiency ATF levels in whole cell lysates 24 hours after exposure to UV. They also use trabectedin, which has reduced cytotoxic action in TCR-deficient cell lines; however, additional DNA damage response mechanisms have been implicated in its action, and thus it can only be used as a complementary assay to RRS to characterize TCR deficiency).

To address this point, we have performed RRS assays on all cell-lines. As expected, all TCR-KO cells lines show a clear and complete RRS defect (Figure 2d).

Minor comments:

(a) What is the reason that the authors use TP53 knock-out cells to construct their isogenic cell lines? As this will also compromise the efficiency of the second NER pathway (GGR) and render the cell more sensitive.

The knockout of ELOF1 is only possible in a TP53^{KO} background, which is also seen for other genes, such as BRCA1. Knockout of TP53 does not affect GGR, as demonstrated by normal UDS in these cells, which is fully dependent on XPC (and all other genes involved in GGR).

As an example of this, we previously compared primary XP-A patient fibroblasts (XP2LD) carrying a homozygous H244R substitution with RPE1 XPA^{KO} cells (that are also TP53 and CSB knockout). We rescued these cells with either XPA^{WT} or XPA^{H244R}. The level of UDS in primary cells or in our H244R rescue cells was very similar. See below, taken from: <https://pubmed.ncbi.nlm.nih.gov/36893274/>

[REDACTED]

(b) Figure 1c, the y-axis label has a spelling mistake (“damge” instead of “damage”).

This has been corrected in the revised manuscript.

(c) Figure 1e and other similar figures in the text. An indication that dots in the western blots represent the UV-treated samples is missing, so it is clearer for the reader.

This has been corrected in the revised manuscript.

(d) Figure 2a. RNAPII indication is missing from the schematic.

This has been corrected in the revised manuscript.

Reviewer #3 (Remarks to the Author)

The present study by van der Meer and Collaborators provide an interesting observation: cells knock out for either CSA or CSB genes display a failure in RNA polymerase II (RNAPII) clearance at the site of UV-induced DNA lesions, likely associated with a lack of RNAPII post-translational modifications. Mutations in either CSA or CSB genes are causative of Cockayne syndrome (CS), the severe Cerebro-oculo-facio-skeletal syndrome (COFS) or the very mild UV sensitive syndrome (UVSS). Failure of RNAPII clearance is not observed in cells defective for other genes encoding proteins involved in DNA repair by nucleotide excision repair (NER) and associated to another pathological condition, the xeroderma pigmentosum. Based on these findings, the Authors suggest that RNAPII clearing defects may underlie the Cockayne syndrome-like neurodegenerative phenotype. While the observations may be interesting and worth of further investigations, the findings are still too preliminary and the conclusions not supported by the patient's clinical spectrum. Overall, the data cannot justify a publication in Nature Communications.

Major concerns:

(1) Defects in CSB or CSA genes are not always associated to neurological defects and/or neurodegeneration. In particular, Horibata K and Colleagues (doi: 10.1073/pnas.0404587101) demonstrated that homozygous mutation in the human genome leading to the loss of CSB protein synthesis (a nonsense mutation in position 77 of CSB) resulted in a UVSS patient with no neurological abnormalities; Nardo T and Colleagues (doi: 10.1073/pnas.0902113106) demonstrated that a missense mutation in position 361 of CSA resulted in failure of transcription-coupled repair (TCR) but not neurological abnormalities or neurodegeneration. These observations are in contrast with the Authors' claim that their findings can explain the Cockayne syndrome-like neurodegenerative phenotype.

We disagree. Both cases of CSB and CSA mutations without Cockayne syndrome (at the time of diagnosis when these individuals were 33 and 15 years, respectively) have puzzled the field for a long time and an explanation is still lacking. One explanation that cannot be excluded is that both cases actually represent late-onset CS. For instance, consider the CS-B patient (KPSX6) described in: Hashimoto, S. et al. *Adult-onset neurological degeneration in a patient with Cockayne syndrome and a null mutation in the CSB gene. J Invest Dermatol 128, 1597-9 (2008)*. The patient was diagnosed with CS at age 47 years, following normal mental development. Neurological abnormalities only became apparent after the age of 47 years, meaning that this patient would have been classified as UV^S, which is indistinguishable from late-onset CS before the onset of symptoms. In our University Medical Center, we also diagnosed a late-onset CS case at age 34 (CS1LD), who has a normal IQ and only exhibited neurological abnormalities recently.

We included several patient-derived cells to evaluate the performance of our RNAPII clearance assays. Normal clearance is detected in WT primary fibroblasts (48BR, or 48BR-hTERT). Fibroblasts from a severe CS-B patient (CS1AN-hTERT) showed no RNAPII clearance. UVSSA-deficient Kps3-hTERT fibroblasts exhibited delayed RNAPII clearance, similar to what we observed in UVSSA knockout RPE1 cells. We also included immortalized fibroblasts (XP8BR-hTERT) from a compound heterozygous XP-D/CS patient with clinical features characteristic of CS. This patient has a frameshift mutation on the paternal allele and a G675R substitution on the maternal allele, both confirmed by Sanger sequencing. Additionally, RRS experiments confirmed a strong TCR defect in these cells. We observed a significant RNAPII clearance defect in XP8BR-hTERT cells, comparable to that in CS-B cells (Fig 5a, b).

As requested, we investigated whether RNAPII clearance is affected in cells from two unusual TCR patients with CSB (UV^S1KO) and CSA (UV^S1VI) mutations, who had no neurodegenerative symptoms at diagnosis (ages 33 and 15). The CS-B patient has an R77X stop codon, and the CS-A patient has a W361C substitution, confirmed by Sanger sequencing. Cells from both patients showed defects in TCR and RNAPII clearance. Rescue of CSA knockout RPE1 cells with CSA^{W361C} failed to restore RNAPII clearance, while CSA^{WT} did. The CSA^{W361C} mutant protein is mislocalized to the cytoplasm and defective in RRS, suggesting that this substitution affects CSA protein folding. We cannot exclude that these unusual cases reflect late-onset CS, as adult-onset CS cases often resemble UV^S patients before symptom onset. Indeed, cells from a late-onset CS case (CS1LD), diagnosed at our University Medical Center at age 34 with a normal IQ, also exhibited RNAPII clearance defects, suggesting that this assay may detect late-onset CS (Fig 5c-e).

However, we also describe in the discussion that:

We find that RNAPII clearance generally correlates well with CS features, but we note that we detect strongly affected clearance in a late-onset CS case (CS1LD), as well as in unusual cases with CSB (UV^S1KO) or CSA (UV^S1VI) mutations, who did not present with a CS neurodegenerative phenotype at the time of diagnosis. It is possible that these individuals may develop neurodegenerative features later in life, as reported for other adult-onset CS cases. It is important to note that our assay requires a high local UV damage load to detect the damaged area for measuring RNAPII clearance. This high damage load may result in an apparent RNAPII clearance defect that would not manifest under a lower, more physiological damage load, potentially leading to a milder phenotype.

Several articles demonstrated that an accumulation of oxidative stress in CS cells, likely caused by additional roles of CSA and CSB outside TCR, is more directly correlated with CS neurodegeneration.

We disagree. The sensitivity of CS cells to oxidative stress is often minimal or absent. A systematic approach to studying this is currently lacking. For instance, Hanawalt and co-workers developed a sensitive Comet-FISH method, which showed that transcribed strand-specific repair of 8-oxoGs in human cells depends not only on CSB and CSA, but also on UVSSA (<https://pubmed.ncbi.nlm.nih.gov/23775797/>). Therefore, the available data suggest that the accumulation of oxidative damage does not necessarily correlate directly with CS neurodegeneration. In fact, patients with mutations in oxidative DNA damage repair due to XRCC1 inactivation exhibit late-onset ataxia (<https://www.nature.com/articles/nature20790>), which is quite distinct from CS. If oxidative damage accumulation were the primary cause, XRCC1 patients would be expected to exhibit symptoms of CS, which they do not.

In addition to these theoretical considerations, we tested the sensitivity of cells to oxidative DNA damage induced by potassium bromate (KBrO₃). We performed cell viability assays on isogenic RPE1 cells exposed to 100–400 μM KBrO₃. As anticipated, XRCC1^{KO} cells showed sensitivity, but we observed no appreciable sensitivity in CSB^{KO} cells or patient-derived CS1AN-hTERT cells compared to wild-type RPE1-hTERT cells. Our findings do not support the idea that CSB-deficient cells are sensitive to oxidative damage in isogenic cell lines (Fig 51c).

We also confirm, as demonstrated by Nardo et al. (2009), that UV^S1VI cells are not sensitive to KBrO₃. It is important to note that their experiments used continuous treatment with very high doses of KBrO₃ (10 or 20 mM), which already caused significant sensitivity in wild-type cells. Under our conditions, we may detect some sensitivity in UV^S1KO cells. However, these cells come from a CS-B patient diagnosed with UV^S, showing that oxidative stress does not correlate with neurodegeneration.

Additionally, since these cells are not isogenic, we believe direct comparisons are difficult. For instance, when we included wild-type VH10-SV cells for comparison with UV^S1KO-SV cells, we found that the wild-type cells were much more sensitive. Furthermore, wild-type VH10-SV cells were significantly more sensitive than wild-type 48BR cells. This highlights the importance of using isogenic cell lines for accurate comparison. Under isogenic conditions, we detected sensitivity for XRCC1^{KO} but not for CSB^{KO}. We therefore decided not to include the data below in the revised manuscript.

Our results in RPE1 cells aligns with results from recent CRISPR screens showing that sgRNA targeting CSA or CSB did not result in sensitivity to oxidative DNA damage induced by potassium bromate (KBrO₃), or methylnitrosoguanidine (MNNG), whereas sgRNA against the essential BER scaffold XRCC1 did (now shown in Fig S1a, b).

(2) CS patients may display neurological defects and/or neurodegeneration. XPA-defective individuals do not display CS-like features, nevertheless they always exhibit neurodegeneration. Considering that XPA knock out cells generated in this study do not show RNAPII clearance defects, it is unlikely that “deficiency in RNAPII processing and prolonged transcription arrests in response to DNA damage, rather than compromised DNA repair, may underlie the Cockayne syndrome-like neurodegenerative phenotype”.

We disagree. The neurological phenotype of XP-A patients is very distinct from the neurodegeneration seen in Cockayne syndrome. Furthermore, XP-A patients do not always exhibit neurodegeneration. Depending on how neurodegeneration is defined, around 25% of XP-A patients show neurodegenerative manifestations. Nonetheless, the reviewer seems to agree by stating that "*XPA-defective individuals do not display CS-like features*," which is precisely our point. The CS-like features, seen only in CS-A and CS-B individuals, correlate at the cellular level with defective RNAPII clearance.

(3) The Authors claim that the appearance of the RNAPII-o upper band upon UV irradiation corresponds to the ubiquitinated form of elongating RNAPII (Fig. 5 and 6). Even though this may be likely, the statement should be proved with specific assays (using anti-ubiquitin antibodies, proteasome inhibitors, specific immunoprecipitations, etc.).

We have demonstrated this before with ubiquitin antibodies (left) or by treating RNAPII after IP from cells with a recombinant ubiquitin-specific protease (USP2; see right), leading to complete loss of this upper band. We have demonstrated specificity of the upper band as the ubiquitylation form of RNAPII. We mapped the site (K1268) and knock-in of K1268R abolishes the upper band. These experiments have been performed and published. Please see: <https://www.sciencedirect.com/science/article/pii/S0092867420301549>

[REDACTED]

(4) While CSB, CSA, UVSSA can be defined TCR factors, XPA, XPG, ERCC1, TFIIH should be named NER factors (as their inactivation impairs both global genome and transcription-coupled repair).

Yes, they are GGR factors, too. ERCC1 is also an ICL repair factor and an SSA factor. Likewise, TFIIH is also a general transcription factor. We introduce them as NER factors in the introduction:

“The collection included cells with genetic inactivation of early TCR factors (CSB, CSA), factors that stabilize the RNAPII-bound TCR complex and recruit TFIIH (ELOF1, UVSSA), and downstream NER factors that act after RNAPII displacement to repair the actual lesions (XPA, XPG, ERCC1) (Fig 2a).”

However, since we study degradation and clearance of RNAPII in these KO cells, the only relevant pathway in this context is TCR. For clarity, we therefore refer to them in this context as TCR factors.

(5) Statistical analyses are missing. Proper experimental controls are missing (negative controls such as cells without DRB treatment). No clear explanation is provided concerning the different type of RNAPII antibodies used (which population of phosphorylated forms of RNAPII are recognized).

Statistical analyses have been added.

Concerning 'proper' controls: Our new assay critically depends on adding DRB to prevent new RNAPII transitioning into elongation. Therefore, performing this assay without DRB is not meaningful. It is not an experimental control, but rather a completely different set-up in which new RNAPII molecules will initiate and encounter the same unrepaired DNA lesions over and over. With DRB, we can track the data of the elongating RNAPII pool without new RNAPII initiating at the same time.

The different RNAPII antibodies are common in the field and used to detect Ser2/5-phosphorylated RNAPII (the elongating RNAPII_o form), or to detect all forms of RNAPII (both RNAPII_o; phosphorylated, as well as RNAPII_a; unphosphorylated). The Leiden team uses (3E10) while the Nagoya team uses H5 to detect Ser2-phosphorylated RNAPII using western blot. The D8L4Y is used to detect all forms of RNAPII using western blot. The ab5095 antibody is used to detect Ser2-phosphorylated RNAPII by immunofluorescence.

Reviewer #1 (Remarks to the Author):

The revisions to the manuscript have addressed my previous concerns and significantly improved the manuscript.

We thank the reviewer for their helpful suggestions.

Reviewer #2 (Remarks to the Author):

Overall, I find this manuscript improved compared to the previous version.

However, the concern regarding novelty remains valid, as the study investigating the same mechanism and reaching similar conclusions was published in early September (Gonzalo-Hansen, et al., NAR) following its initial release as a preprint a year ago. Given that this manuscript is largely centred on this mechanism, I believe it lacks the novelty necessary for consideration for publication in Nature Communications.

We have substantially revised our manuscript to delve deeper into mechanisms that contribute to RNAPII clearance. Our new findings establish a hierarchical program in which CSB/CRL4^{CSA}-mediated ubiquitylation initiates RNAPII processing, TFIIH/XPD helicase activity provides the main clearance mechanism, and VCP-dependent extraction acts as a backup when TFIIH fails. This mechanistic framework explains how cells resolve DNA lesion-arrested RNAPII during normal and compromised TCR.

Below are a few comments that need to be addressed.

Regarding my previous comment on chromatin fraction and RNAPII S2 detection, I agree with the authors that this elongating form of RNAPII is exclusively associated with chromatin. Since the authors use various antibodies to detect RNAPII phosphorylation status—some of which may lack precise specificity for RNAPII S2 versus S5 phosphorylation—they should indicate in the text or figure legends which antibody is used for each specific experiment. While Supplementary Table 5 lists the different RNAPII antibodies used for western blotting, the authors do not specify in the text which antibodies were applied in each experiment.

We indicate this now in the revised manuscript's figure legends.

A couple of points that I unfortunately missed in the previous revision are the following:

In Figure 1e, the authors report that ‘The antibody against the N-terminal domain of RNAPII detects both initiating (RNAPIIa) and the elongating (RNAPIIo) forms (middle blot). Note that RNAPII-S2 and RNAPIIo represent the same pool of phosphorylated RNAPII.’

Could the authors clarify why RNAPII-S5 is not detectable in their extracts when using an antibody that targets the N-terminus of the RNAPII RPB1 subunit, which should theoretically recognize both non-phosphorylated and all the phosphorylated states of RNAPII? Please correct the text accordingly.

We detect two distinct bands with the N-terminal antibody corresponding to initiating (RNAPIIa) and elongating (RNAPIIo). The upper RNAPIIo band is recognized by both RNAPII-S2P and RNAPII-S5P antibodies. The upper RNAPII-S5P bands we detect disappears after DRB and UV. However, the lower RNAPII-S5S, reflecting promoter-proximally paused RNAPII persistent. This is not really a clear single band, but rather a smear. The N-terminal antibody against RNAPII may also show such a smear in between the RNAPIIa and RNAPIIo bands. We now mention the following in the revised manuscript:

‘Staining with an RNAPII-S5P antibody revealed the loss of a high-molecular weight band, corresponding to the RNAPII-S2P and RNAPIIo species, consistent with the elongating, hyperphosphorylated form of RNAPII. In contrast, a lower RNAPII-S5P band persisted, migrating between the unphosphorylated (RNAPIIa) and hyperphosphorylated (RNAPIIo) bands, and representing promoter-proximally paused RNAPII, which persists when cells are treated with DRB’

In a related context, the authors state that ‘DRB prevents new RNAPII complexes from transitioning from the initiation form (RNAPIIa) to the hyperphosphorylated elongation form (RNAPIIo/S2)’. Do the authors believe that DRB inhibits the transition from RNAPIIa to RNAPII-S2? This is established knowledge and should be accurately described and represented. Please revise the diagrams in Supplementary Figure 2 to reflect this accordingly.

We have revised the text and figures accordingly.

Reviewer #3, #4, #5 (Remarks to the Author):

We have strengthened and expanded the molecular data and removed the focus on the patients. We therefore do not respond to points raised by reviewers #3, #4 and #5, which were exclusively about the part of the paper we have now removed.

REVIEWERS' COMMENTS

Reviewer #1 (Remarks to the Author):

The revised manuscript by Van der Meer et al., entitled: "Hierarchical mechanisms control the clearance of DNA lesion–stalled RNA polymerase II" is significantly improved by the inclusion of new data showing that the TFIIH subunit XPD plays an important role in RNA polymerase II removal following stalling at UV damage. The authors now suggest a mechanism in which XPD plays a major role in removing lesion-stalled polymerase and VCP/p97 segregase plays a backup role. While I thought the original version of the manuscript was acceptable, the new version is that much better. I had only one minor suggestion:

We thank the reviewer for their time and constructive comments, and for appreciating the previous and current versions of our manuscript.

1. On line 305/306, the text, "separate-of-function mutant", I think should be: "separation-of-function mutant".

We corrected this in the current manuscript.

Reviewer #2 (Remarks to the Author):

The current revised manuscript has addressed adequately my concerns.

We thank the reviewer for their supportive feedback.

Reviewer #6 (Remarks to the Author):

This manuscript investigates how RNA polymerase II (RNAPII) is cleared from sites of transcription-blocking DNA lesions and proposes a hierarchical model in which TFIIH-driven clearance depends on stimulation by downstream nucleotide excision repair (NER) factors, while, in their absence, VCP (p97) mediates RNAPII removal. The authors build on their previously described DRB run-off assay to monitor RNAPII clearance kinetics in cells deficient for various TCR and NER components.

The study is technically strong and addresses an important question in transcription-coupled DNA repair (TCR). It provides the first systematic analysis of the contribution of downstream NER factors (TFIIH, XPA, XPG, ERCC1-XPF) to RNAPII removal, extending beyond earlier reports that focused primarily on upstream TCR components (CSB, CSA, UVSSA) (NAR, 2024, <https://doi.org/10.1093/nar/gkae618>; Nature Communications, 2024, <https://doi.org/10.1038/s41467-024-51463-x>; NAR, 2023, <https://doi.org/10.1093/nar/gkad008>). The data are well organized and of high quality. However, while the work provides valuable insights into the Pol II clearance process, a few points need to be clarified.

Major point

1. The manuscript proposes that downstream NER factors (XPA, XPG, ERCC1–XPF) stimulate XPD helicase to promote RNAPII clearance, implying that clearance occurs after the recruitment of these factors. However, a previous study (van der Weegen et al., Nat Commun 2020, <https://doi.org/10.1038/s41467-020-15903-8>) reported that although CS proteins and TFIIH readily assembled with RNAPII after UV irradiation, downstream repair proteins such as XPA and ERCC1–XPF could not be detected by immunoprecipitation. The authors should discuss this apparent contradiction.

2. The manuscript describes XPD helicase as the primary driver of RNAPII removal, which may be a somewhat over-interpretation of current data. While XPD helicase activity is clearly important for NER and could contribute to local DNA unwinding or remodeling around stalled RNAPII, direct solid biochemical evidence that XPD actively drives RNAPII displacement in TCR is still lacking. The authors may wish to tune down the statement and acknowledge that this remains a working hypothesis to be further tested in future mechanistic studies.

3. The manuscript states that downstream NER factors stimulate XPD helicase to promote RNAPII clearance. However, the authors may wish to clarify this point, as the direct evidence supporting this mechanism appears

limited. A more open and cautious wording would be advisable when describing this relationship. Points 1, 2 and 3 are addressed in this new paragraph at the end of the results section:

“Although XPA, XPG, and ERCC1-XPF have not previously been detected in the same complex as damage-stalled RNAPII, our findings imply that these downstream NER factors are recruited to TFIIH while TFIIH is bound to RNAPII, presumably to promote XPD helicase activity. A possible explanation for why current approaches have not detected this interaction is that, upon NER factor binding, TFIIH rapidly adopts its active repair conformation, and RNAPII is cleared, resulting in a very short-lived interaction that is difficult to capture. Future studies should aim to capture this interaction more accurately and assess its effects on TFIIH helicase activity and RNAPII clearance, for example, using proximity-labeling and biochemical approaches.”

4. Previous studies have reported the involvement of upstream TCR components (CSB, CSA, UVSSA) in RNAPII clearance (NAR, 2024, <https://doi.org/10.1093/nar/gkae618> ; Nature Communications, 2024, <https://doi.org/10.1038/s41467-024-51463-x> ; NAR, 2023, <https://doi.org/10.1093/nar/gkad008>). The present study appears to report related findings, and it may be helpful for the authors to discuss their results in the context of these previous reports to provide a more balanced and integrated perspective.

The third paragraph of the introduction and the second paragraph of the discussion already discuss the studies that the reviewer mentioned extensively. We have added an additional comment on page 9: “These results align well with the proposed role of VCP in repair-independent eviction of lesion-stalled RNAPII in the absence of UVSSA, detected by Damage-seq and PADD-seq.”

5. The manuscript first states that “To confirm that the loss of hyperphosphorylated RNAPII-S2P indeed reflects proteasome-mediated degradation, we included the proteasome inhibitor MG132 in our experiments. Both local and global DRB run-off methods showed that inhibiting the proteasome abrogated the clearance and degradation of damage-stalled RNAPII from chromatin (Figure 1h–k)” (page 5–6, lines 138–142). Later, however, it is noted that “Treatment of cells with VCPi led to a small but nonsignificant effect on RNAPII clearance in WT cells, indicating that VCP-mediated chromatin extraction does not play a significant role in TCR-proficient cells (Figure 4a, Figure S5a)” (page 8, lines 275–278).

As VCP/p97 is generally considered an essential cofactor in proteasome-mediated degradation, the authors may wish to clarify how these observations could be reconciled, or whether they reflect distinct experimental conditions or mechanisms of RNAPII clearance.

We note that VCP extracts chromatin-bound ubiquitylated proteins and shuttles them to the proteasome. Proteasomal degradation does not always require VCP. Therefore, VCPi inhibits only a subset of proteasomal degradation processes.

A caveat with MG132 treatment is that this is known to deplete the free ubiquitin pool, thereby impairing ubiquitylation more broadly. To show this, we performed immunoprecipitation of RNAPII-S2P after MG132 treatment which revealed a marked reduction in RNAPII-S2P ubiquitylation, indicating that the observed defects in RNAPII-S2P clearance and degradation likely reflect combined effects of proteasome inhibition and impaired ubiquitylation. This new data is now shown in Supplementary Fig. 1b.

6. The manuscript focuses on VCP/p97-dependent RNAPII clearance but does not examine how this process relates to proteasome-mediated degradation. Given that VCP/p97 typically acts upstream of the proteasome to extract ubiquitylated substrates from chromatin, it remains unclear whether RNAPII removed in this manner is ultimately degraded or recycled. Previous work (Nature Communications, 2024, <https://doi.org/10.1038/s41467-024-51463-x>) has directly linked proteasome activity to RNAPII degradation under UVSSA-deficient conditions. The authors may wish to discuss how their findings align or differ from these results to clarify the mechanistic relationship between VCP and the proteasome.

We thank the reviewer for this comment. Indeed, VCP/p97 acts upstream of the proteasome to extract ubiquitylated substrates from chromatin. In the comment point #5, we show that using MG132 is not the optimal way to study RNAPII clearance and proteasomal degradation. Therefore, in Figure 4a-c, we analyzed RNAPII clearance, degradation, and ubiquitylation after VCP inhibition, showing that UVSSA-deficient cells fully rely on VCP for RNAPII clearance (4a), proteasomal degradation (4b), and accumulate ubiquitylated RNAPII on chromatin (4c). We accordingly modified that text to emphasize these findings.

Minor point

1. On page 13, line 488, the text reads “(Figure S5d)” in the sentence “Superimposing an STK19-XPD AlphaFold model onto experimental cryo-EM structures of the RNAPII-bound TCR complex, including STK19, positions XPD very close to the RPB1K1268 ubiquitylation site (Figure S5d)”. This appears to be a typo — the correct reference should be “(Figure S6d)”.

This has been corrected.

2. The manuscript employs the UVSSA Δ TFIIH mutant but does not cite a reference describing its design or the deleted region. The authors should indicate the source or provide details of the deleted amino acid sequence to ensure clarity and reproducibility.

We added the missing information and the reference to the results and method sections.

Reviewer #7 (Remarks to the Author):
